# ON THE FEASIBILITY OF CROSS-TASK TRANSFER WITH MODEL-BASED REINFORCEMENT LEARNING

**Yifan Xu**,* **Nicklas Hansen**,* **Zirui Wang, Yung-Chieh Chan, Hao Su, Zhuowen Tu**
University of California, San Diego
{yix081, nihansen, ziw029, ychan, has168, ztu}@ucsd.edu

## ABSTRACT

Reinforcement Learning (RL) algorithms can solve challenging control problems directly from image observations, but they often require millions of environment interactions to do so. Recently, model-based RL algorithms have greatly improved sample-efficiency by concurrently learning an internal model of the world, and supplementing real environment interactions with imagined rollouts for policy improvement. However, learning an effective model of the world from scratch is challenging, and in stark contrast to humans that rely heavily on world understanding and visual cues for learning new skills. In this work, we investigate whether internal models learned by modern model-based RL algorithms can be leveraged to solve new, distinctly different tasks faster. We propose Model-Based **Cross**-Task **Tra**nsfer (**XTRA**), a framework for sample-efficient online RL with scalable pretraining and finetuning of learned world models. By offline multi-task pretraining and online cross-task finetuning, we achieve substantial improvements over a baseline trained from scratch; we improve mean performance of model-based algorithm EfficientZero by 23%, and by as much as 71% in some instances.

## 1 INTRODUCTION

Reinforcement Learning (RL) has achieved great feats across a wide range of areas, most notably game-playing (Mnih et al., 2013; Silver et al., 2016; Berner et al., 2019; Cobbe et al., 2020). However, traditional RL algorithms often suffer from poor sample-efficiency and require millions (or even billions) of environment interactions to solve tasks – especially when learning from high-dimensional observations such as images. This is in stark contrast to humans that have a remarkable ability to quickly learn new skills despite very limited exposure (Dubey et al., 2018). In an effort to reliably benchmark and improve the sample-efficiency of image-based RL across a variety of problems, the Arcade Learning Environment (ALE; (Bellemare et al., 2013)) has become a long-standing challenge for RL. This task suite has given rise to numerous successful and increasingly sample-efficient algorithms (Mnih et al., 2013; Badia et al., 2020; Kaiser et al., 2020; Schrittwieser et al., 2020; Kostrikov et al., 2021; Hafner et al., 2021; Ye et al., 2021), notably most of which are model-based, *i.e.*, they learn a *model* of the environment (Ha & Schmidhuber, 2018).

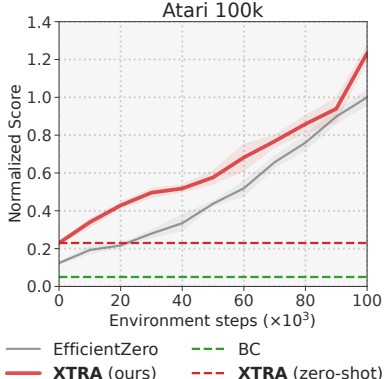

*Figure 1*. **Atari100k score**, normalized by mean EfficientZero performance at 100k environment steps across 10 games. Mean of 5 seeds; shaded area indicates 95% CIs.

Most recently, EfficientZero Ye et al. (2021), a model-based RL algorithm, has demonstrated impressive sample-efficiency, surpassing human-level performance with as little as 2 hours of real-time game play in select Atari 2600 games from the ALE. This achievement is attributed, in part, to the algorithm concurrently learning an internal *model* of the environment from interaction, and using the learned model to *imagine* (simulate) further interactions for planning and policy improvement, thus reducing reliance on real environment interactions for skill acquisition. However, current RL algorithms, including EfficientZero, are still predominantly assumed to learn both perception, model, and skills

*Equal contribution. Project page: https://nicklashansen.github.io/xtra.

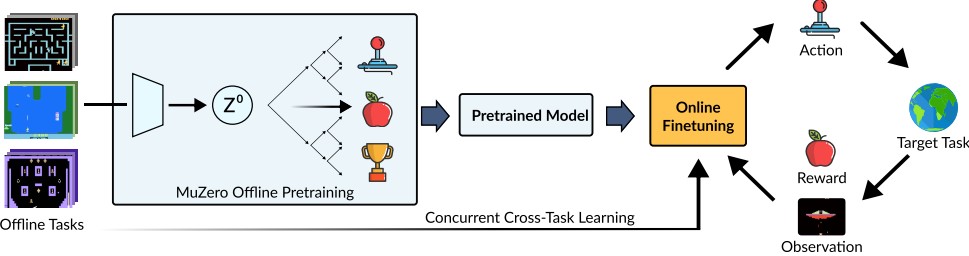

**Figure 2.** Model-Based **Cross**-Task **Tra**nsfer (**XTRA**): a sample-efficient online RL framework with scalable pretraining and finetuning of learned world models using auxiliary data from offline tasks.

*tabula rasa* (from scratch) for each new task. Conversely, humans rely heavily on prior knowledge and visual cues when learning new skills – a study found that human players easily identify visual cues about game mechanics when exposed to a new game, and that human performance is severely degraded if such cues are removed or conflict with prior experiences (Dubey et al., 2018).

In related areas such as computer vision and natural language processing, large-scale unsupervised/self-supervised/supervised pretraining on large-scale datasets (Devlin et al., 2019; Brown et al., 2020; Li et al., 2022; Radford et al., 2021; Chowdhery et al., 2022) has emerged as a powerful framework for solving numerous downstream tasks with few samples (Alayrac et al., 2022). This pretraining paradigm has recently been extended to visuo-motor control in various forms, *e.g.*, by leveraging *frozen* (no finetuning) pretrained representations (Xiao et al., 2022; Parisi et al., 2022) or by finetuning in a supervised setting (Reed et al., 2022; Lee et al., 2022). However, the success of finetuning for *online RL* has mostly been limited to same-task initialization of model-free policies from offline datasets (Wang et al., 2022; Zheng et al., 2022), or adapting policies to novel instances of a given task (Mishra et al., 2017; Julian et al., 2020; Hansen et al., 2021a), with prior work citing high-variance objectives and catastrophical forgetting as the main obstacles to finetuning representations with RL (Bodnar et al., 2020; Xiao et al., 2022).

In this work, we explore whether such positive transfer can be induced with current model-based RL algorithms in an *online* RL setting, and across *markedly distinct* tasks. Specifically, we seek to answer the following questions: *when* and *how* can a model-based RL algorithm such as EfficientZero benefit from pretraining on a diverse set of tasks? We base our experiments on the ALE due to cues that are easily identifiable to humans despite great diversity in tasks, and identify two key ingredients – cross-task finetuning and task alignment – for model-based adaptation that improve sample-efficiency substantially compared to models learned tabula rasa. In comparison, we find that a naïve treatment of the finetuning procedure as commonly used in supervised learning (Pan & Yang, 2010; Doersch et al., 2015; He et al., 2020; Reed et al., 2022; Lee et al., 2022) is found to be unsuccessful or outright *harmful* in an RL context.

Based on our findings, we propose Model-Based **Cross**-Task **Tra**nsfer (**XTRA**), a framework for sample-efficient online RL with scalable pretraining and finetuning of learned world models using extra, auxiliary data from other tasks (see Figure 2). Concretely, our framework consists of two stages: *(i) offline multi-task pretraining* of a world model on an offline dataset from $m$ diverse tasks, a *(ii) finetuning* stage where the world model is jointly finetuned on a *target task* in addition to $m$ offline tasks. By leveraging offline data both in pretraining and finetuning, XTRA overcomes the challenges of catastrophical forgetting. To prevent harmful interference from certain offline tasks, we adaptively re-weight gradient contributions in unsupervised manner based on similarity to target task.

We evaluate our method and a set of strong baselines extensively across 14 Atari 2600 games from the Atari100k benchmark (Kaiser et al., 2020) that require algorithms to be extremely sample-efficient. From Figure 1 and Table 1, we observe that XTRA improves sample-efficiency substantially across most tasks, improving mean and median performance of EfficientZero by 23% and 25%, respectively.

## 2 BACKGROUND

**Problem setting.** We model image-based agent-environment interaction as an episodic Partially Observable Markov Decision Process (POMDP; Kaelbling et al. (1998)) defined by the tuple $\mathcal{M} =$

$\langle \mathcal{O}, \mathcal{A}, \mathcal{P}, \rho, r, \gamma \rangle$, where $\mathcal{O}$ is the observation space (pixels), $\mathcal{A}$ is the action space, $\mathcal{P} \colon \mathcal{O} \times \mathcal{A} \mapsto \mathcal{O}$ is a transition function, $\rho$ is the initial state distribution, $r \colon \mathcal{O} \times \mathcal{A} \mapsto \mathbb{R}$ is a scalar reward function, and $\gamma \in [0, 1)$ is a discount factor. As is standard practice in ALE (Bellemare et al., 2013), we convert $\mathcal{M}$ to a fully observable Markov Decision Process (MDP; Bellman (1957)) by approximating state $\mathbf{s}_t \in \mathcal{S}$ at time $t$ as a stack of frames $\mathbf{s}_t \doteq \{o_t, o_{t-1}, o_{t-2}, \dots\}$ where $o \in \mathcal{O}$ (Mnih et al., 2013), and redefine $\mathcal{P}, \rho, r$ to be functions of $\mathbf{s}$. Our goal is then to find a (neural) policy $\pi_\theta(\mathbf{a}|\mathbf{s})$ parameterized by $\theta$ that maximizes discounted return $\mathbb{E}_{\pi_\theta}[\sum_{t=1}^{t} \gamma_t r(\mathbf{s}_t, \mathbf{a}_t)]$ where $\mathbf{a}_t \sim \pi_\theta(\mathbf{a}|\mathbf{s})$, $\mathbf{s}_t \sim \mathcal{P}(\mathbf{s}_t, \mathbf{a}_t), \mathbf{s}_0 \sim \rho$, and $T$ is the episode horizon. For clarity, we denote all parameterization by $\theta$ throughout this work. To obtain a good policy from minimal environment interaction, we learn a *"world model"* from interaction data and use the learned model for action search. Define $\mathcal{M}$ as the *target task* that we aim to solve. Then, we seek to first obtain a good parameter initialization $\theta$ that allows us to solve task $\mathcal{M}$ using fewer interactions (samples) than training from scratch, *i.e.*, we wish to improve the *sample-efficiency* of online RL. We do so by first pretraining the model on an *offline* (fixed) dataset that consists of transitions $(\mathbf{s}, \mathbf{a}, r, \mathbf{s}')$ collected by unknown behavior policies in $m$ environments $\{\tilde{\mathcal{M}}^i \mid \tilde{\mathcal{M}}^i \neq \mathcal{M}, 1 \leq i \leq m\}$, and then *finetune* the model by online interaction on the target task.

**EfficientZero** (Ye et al., 2021) is a model-based RL algorithm based on MuZero (Schrittwieser et al., 2020) that learns a discrete-action latent dynamics model from environment interactions, and selects actions by lookahead via Monte Carlo Tree Search (MCTS; (Abramson, 1987; Coulom, 2006; Silver et al., 2016)) in the latent space of the model. Figure 3 provides an overview of the three main components of the MuZero algorithm: a representation (encoder) $h_\theta$, a dynamics (transition) function $g_\theta$, and a prediction head $f_\theta$. Given an observed state $\mathbf{s}_t$, EfficientZero projects the state to a latent representation $\mathbf{z}_t = h_\theta(\mathbf{s}_t)$, and predicts future latent states $\mathbf{z}_{t+1}$ and instantaneous rewards $\hat{r}_t$ using an action-conditional latent dynamics function $\mathbf{z}_{t+1}, \hat{r}_t = g_\theta(\mathbf{z}_t, \mathbf{a}_t)$. For each latent state, a prediction network $f_\theta$ estimates a probability

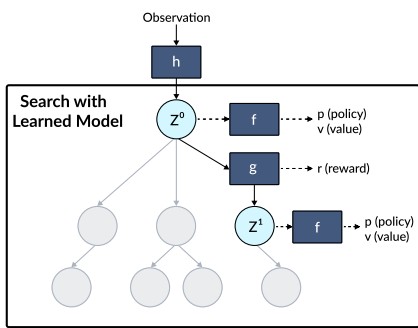

*Figure 3.* **MuZero/EfficientZero** combines MCTS with a learned representation network ($h$), latent dynamics function ($g$), and prediction head ($f$).

distribution $\hat{p}$ over (valid) actions $\mathbf{a} \in \mathcal{A}$, as well as the expected state value $\hat{v}$ of the given state, *i.e.*, $\hat{v}_t, \hat{p}_t = f_\theta(\mathbf{z}_t)$. Intuitively, $h_\theta$ and $g_\theta$ allow EfficientZero to search for actions entirely in its latent space before executing actions in the real environment, and $f_\theta$ predicts quantities that help guide the search towards high-return action sequences. Concretely, $\hat{v}$ provides a return estimate for nodes at the lookahead horizon (as opposed to truncating the cumulative sum of expected rewards) and $\hat{p}$ provides an action distribution prior that helps guide the search. We describe EfficientZero's learning objective between model prediction ($\hat{p}, \hat{v}, \hat{r}$) and quantity targets ($\pi, z, u$) in Appendix A. EfficientZero improves the sample-efficiency of MuZero by introducing additional auxiliary losses during training. We adopt EfficientZero as our backbone model and learning algorithm, but emphasize that our framework is applicable to most model-based algorithms, including those for continuous action spaces (Hafner et al., 2019a; Hansen et al., 2022a).

## 3 MODEL-BASED CROSS-TASK TRANSFER

We propose Model-Based **Cross**-Task **Tra**nsfer (**XTRA**), a two-stage framework for offline multi-task pretraining and cross-task transfer of learned world models by finetuning with online RL. Specifically, we first pretrain a world model on offline data from a set of diverse pretraining tasks, and then iteratively finetune the pretrained model on data from a *target* task collected by online interaction. In the following, we introduce each of the two stages – pretraining and finetuning – in detail.

### 3.1 OFFLINE MULTI-TASK PRETRAINING

In this stage, we aim to learn a single world model with general perceptive and dynamics priors across a diverse set of offline tasks. We emphasize that the goal of pretraining is not to obtain a truly generalist agent, but rather to learn a good initialization for finetuning to unseen tasks. Learning a single RL agent for a diverse set of tasks is however difficult in practice, which is only exacerbated by extrapolation errors due to the offline RL setting (Kumar et al., 2020). To address such a challenge,

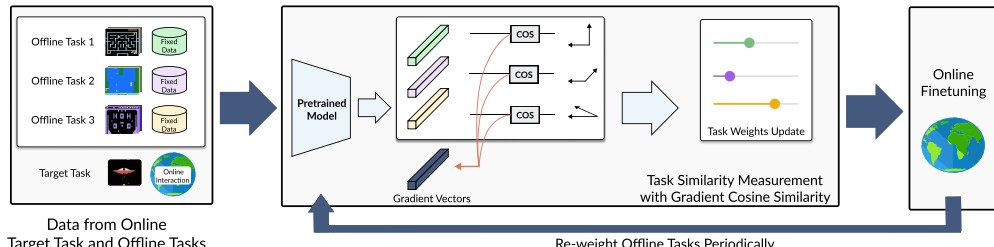

**Figure 4.** Illustration of our proposed **Concurrent Cross-Task Learning** strategy, where we selectively include a subset of the available pretraining tasks while finetuning on a target task.

we propose to pretrain the model following a *student-teacher* training setup in the same spirit to DQN multi-task policy distillation in Rusu et al. (2016) and Actor-Mimic in Parisotto et al. (2016), where *teacher* models are trained separately by offline RL for each task, and then distilled into a single multi-task model using a novel instantiation of the MuZero Reanalyze (Schrittwieser et al., 2021).

For each pretraining task we assume access to a fixed dataset $\{\tilde{\mathcal{D}}^i \,|1 \le i \le m\}$ that consists of trajectories from an unknown (and potentially sub-optimal) behavior policy. Importantly, we do *not* make any assumptions about the quality or the source of trajectories in the dataset, *i.e.*, we do not assume datasets to consist of expert trajectories. We first train individual EfficientZero *teacher* models on each dataset for a fixed number of iterations in a single-task (offline) RL setting, resulting in $m$ *teacher* models $\{\tilde{\pi}_\psi^i \,|1 \le i \le m\}$. After training, we store the model predictions, $(\hat{p}, \hat{v})$, from each *teacher* model $\tilde{\pi}_\psi^i$ together with environment reward $u$ as the student's quantity targets $(\pi, z, u)$ respectively for a given game $\tilde{\mathcal{M}}^i$ (see Appendix A for the definition of each quantity). Next, we learn a *multi-task student* model $\pi_\theta$ by distilling the task-specific teachers into a single model via these quantity targets. Specifically, we optimize the student policy by sampling data uniformly from all pretraining tasks, and use value/policy targets generated by their respective teacher models rather than bootstrapping from student predictions as commonly done in the (single-task) MuZero Reanalyze algorithm. This step can be seen as learning multiple tasks simultaneously with direct supervision by distilling predictions from multiple teachers into a single model. Empirically, we find this to be a key component in scaling up the number of pretraining tasks. Although teacher models may not be optimal depending on the provided offline datasets, we find that they provide stable (due to fixed parameters during distillation) targets of sufficiently good quality. The simpler alternative – training a multi-task model on all $m$ pretraining tasks simultaneously using RL is found to not scale beyond a couple of tasks in practice, as we will demonstrate our experiments in Appendix C. After distilling a multi-task student model, we now have a single set of pretrained parameters that can be used for finetuning to a variety of tasks via online interaction, which we introduce in the following section.

### 3.2 ONLINE FINETUNING ON A TARGET TASK

In this stage, we iteratively interact with a target task (environment) to collect interaction data, and finetune the pretrained model on data from the target task. However, we empirically observe that directly finetuning a pretrained model often leads to poor performance on the target task due to *catastrophical forgetting*. Specifically, the initial sub-optimal data collected from the target task can cause a large perturbation in the original pretrained model parameters, ultimately erasing inductive priors learned during pretraining. To overcome this challenge, we propose a concurrent cross-task learning strategy: we retain offline data from the pretraining stage, and concurrently finetune the model on both data from the target task, as well as data from the pretraining tasks. While this procedure addresses catastrophical forgetting, interference between the target task and certain pretraining tasks can be harmful for the sample-efficiency in online RL. As a solution, gradient contributions from offline tasks are periodically re-weighted in an unsupervised manner based on their similarity to the target task. Figure 4 shows the specific concurrent cross-task learning procedure for target task finetuning in our framework.

At each training step $t$, we jointly optimize the target online task $\mathcal{M}$ and $m$ offline (auxiliary) tasks $\{\tilde{\mathcal{M}}^i \,|\tilde{\mathcal{M}}^i \ne \mathcal{M}, 1 \le i \le m\}$ that were used during the *offline multi-task pretraining* stage. Our online finetuning objective is defined as:

$$\mathcal{L}_t^{\text{adapt}}(\theta) = \mathcal{L}_t^{\text{ez}}(\mathcal{M}) + \sum_{i=1}^{m} \eta^i \mathcal{L}_t^{\text{ez}}(\tilde{\mathcal{M}}^i) \tag{1}$$

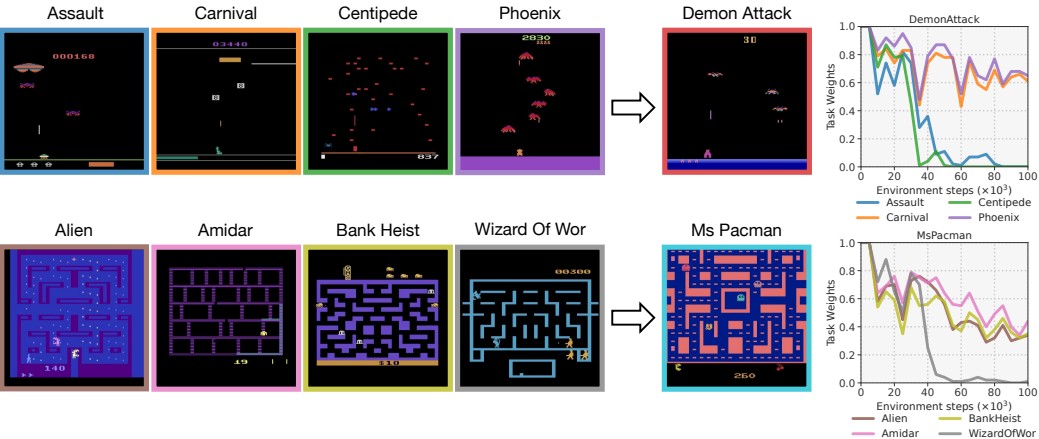

*(a)* Cross-Task Transfer from 4 offline games (left) to 1 target game (right).      *(b)* Task weights.

*Figure 5.* **Visualization of Concurrent Cross-Task Learning.** (*left*) the model adapts to the online target game while concurrently learns 4 offline games. (*right*) the figure shows the task weights of the 4 offline games that are periodically recomputed based on their gradient similarity to the target games (DemonAttack and MsPacman).

where $\mathcal{L}^{\text{ez}}$ is the ordinary (single-task) EfficientZero objective (see Appendix A), and $\eta^i$ are dynamically (and independently) updated task weights for each of the $m$ pretraining tasks. The target task loss term maintains a constant task weight of $1$. During online finetuning, we use distillation targets from teachers obtained from each pretraining game, and use MuZero Reanalyze to compute targets for the target task for which we have no teacher available.

In order to dynamically re-weight task weights $\eta^i$ throughout the training process, we break down the total number of environment steps (*i.e.*, 100k in our experiments) into even $T$-step cycles (intervals). Within each cycle, we spend first $N$-steps to compute an updated $\eta^i$ corresponding to each offline task $\tilde{\mathcal{M}}^i$. The new $\eta^i$ will then be fixed during the remaining $T - N$ steps in the current cycle and the first $N$ steps in the next cycle. We dynamically assign the task weights by measuring the "relevance" between each offline task $\tilde{\mathcal{M}}^i$ and the (online) target task $\mathcal{M}$. Inspired by the conflicting gradients measurement for multi-task learning in Yu et al. (2020), we compute the cosine similarity between loss gradients $\tilde{\mathcal{G}}_n^i$ from $\mathcal{L}_n^{\text{ez}}(\tilde{\mathcal{M}}^i)$ and $\mathcal{G}_n$ from $\mathcal{L}_n^{\text{ez}}(\mathcal{M})$ given by

$$\text{Sim}(\tilde{\mathcal{M}}^i, \mathcal{M}) = \frac{\tilde{\mathcal{G}}_n^i \cdot \mathcal{G}_n}{\|\tilde{\mathcal{G}}_n^i\| \|\mathcal{G}_n\|} . \tag{2}$$

Within the $N$-step update, we maintain a task-specific counter $s^i$ and the new task weights $\eta^i$ can be reset by $\eta^i = \frac{s^i}{N}$ at the beginning of each every $T$-cycle. The procedure for obtaining $s^i$ is described in Appendix B. Concretely, $\text{Sim}(\tilde{\mathcal{M}}^i, \mathcal{M})$ measures the angle between two task gradients $\tilde{\mathcal{G}}_n^i$ and $\mathcal{G}_n$. Intuitively, we aim to (approximately) prevent gradient contributions from the offline tasks from conflicting with the gradient update direction for the target task by regulating offline tasks objectives with task weights $\eta$. While re-weighting task weights at every gradient update would result in the least amount of conflicting gradients, it is prohibitively costly to do so in practice. However, we empirically find the cosine similarity of task gradients to be strongly correlated in time, *i.e.*, the cosine similarity does not change much between consecutive gradient steps. By instead updating task weights every $N$ steps, our proposed technique mitigates gradient conflicts at a negligible computational cost in contrast to the compute-intensive gradient modification method proposed in Yu et al. (2020). Figure 5 shows adjustments to task weights during finetuning for each of two distinct sets of pretraining and target tasks.

# 4 EXPERIMENTS

We evaluate our method and baselines on **14** tasks from the limited-interaction Atari100k benchmark (Kaiser et al., 2020) where only 100k environment steps are permitted. We provide an implementation of our method at `https://nicklashansen.github.io/xtra`. We seek to answer:

- How does our proposed framework compare to alternative pretraining and online RL approaches with *limited* online interaction from the target task?

*Table 1.* **Atari100k benchmark results (*similar* pretraining tasks).** Methods are evaluated at 100k environment steps. For each game, XTRA is first pretrained on all 4 other games from the same category. Our main result is  highlighted . We also include three ablations that remove *(i)* cross-task optimization in finetuning (only online RL), *(ii)* the pretraining stage (random initialization), and *(iii)* task re-weighting (constant weights of 1). We also include zero-shot performance of our method for target tasks in comparison to behavioral cloning. Mean of 5 seeds and 32 evaluation episodes.

| Category | Game | BC (finetuned) | Efficient Zero | Efficient Zero-L | XTRA (Ours) | Ablations (XTRA) | | | Zero-Shot | |
| | | | | | | w.o. cross-task | w.o. pretraining | w.o. task weights | BC | XTRA (Ours) |
|---|---|---|---|---|---|---|---|---|---|---|
| ***Shooter*** | Assault | 838.4 | 1027.1 | 1041.6 | **1294.6** | 1246.4 | 1257.5 | 1164.2 | 0.0 | 92.8 |
| | Carnival | 1952.4 | 3022.1 | 2784.3 | **3860.9** | 3544.4 | 2370.0 | 3071.6 | 93.75 | 719.3 |
| | Centipede | 1814.1 | 3322.7 | 2750.7 | 5681.4 | 3833.2 | **6322.7** | 5484.1 | 162.2 | 1206.8 |
| | Demon Attack | 825.5 | 11523.0 | 4691.0 | 14140.9 | 6381.5 | 9486.8 | **51045.9** | 73.8 | 113.6 |
| | Phoenix | 427.6 | 10954.9 | 3071.0 | 14579.8 | 10797.3 | 9010.6 | **22873.9** | 0.0 | 8073.4 |
| | Mean Improvement | 0.42 | 1.00 | 0.69 | 1.36 | 1.02 | 1.11 | **2.06** | 0.02 | 0.29 |
| | Median Improvement | 0.55 | 1.00 | 0.83 | 1.28 | 1.15 | 0.82 | **1.65** | 0.01 | 0.24 |
| ***Maze*** | Alien | 152.9 | 695.0 | 641.5 | **954.8** | 722.8 | 703.6 | 633.6 | 108.1 | 294.1 |
| | Amidar | 25.5 | 109.7 | 84.2 | 90.2 | **121.8** | 70.8 | 69.7 | 0.0 | 5.2 |
| | Bank Heist | 178.8 | 246.1 | 244.5 | **304.9** | 280.1 | 225.1 | 261.4 | 0.0 | 7.3 |
| | Ms Pacman | 550.0 | 1281.4 | 1172.8 | **1459.7** | 1011.1 | 1122.6 | 809.2 | 147.6 | 448.9 |
| | Wizard Of Wor | 163.8 | 1033.1 | 928.8 | 985.0 | **1246.1** | 654.4 | 263.5 | 100.0 | 9.4 |
| | Mean Improvement | 0.35 | 1.00 | 0.90 | **1.11** | 1.06 | 0.82 | 0.70 | 0.07 | 0.17 |
| | Median Improvement | 0.23 | 1.00 | 0.92 | **1.14** | 1.11 | 0.88 | 0.64 | 0.10 | 0.05 |
| ***Overall*** | Mean Improvement | 0.39 | 1.00 | 0.79 | 1.23 | 1.04 | 0.96 | **1.38** | 0.05 | 0.23 |
| | Median Improvement | 0.33 | 1.00 | 0.91 | **1.25** | 1.12 | 0.85 | 1.04 | 0.02 | 0.16 |

- How do the individual components of our framework influence its success?
- When can we empirically expect finetuning to be successful?

**Experimental setup.** We base our architecture and backbone learning algorithm on EfficientZero (Ye et al., 2021) and focus our efforts on the pretraining and finetuning aspects of our problem setting. We consider EfficientZero with two different network sizes to better position our results: *(i)* the same network architecture as in the original EfficientZero implementation which we simply refer to as **EfficientZero**, and *(ii)* a larger variant with 4 times more parameters in the representation network (denoted **EfficientZero-L**). We use the EfficientZero-L variant as the default network for our framework through our experiments, unless stated otherwise. However, we find that our EfficientZero baseline generally does not benefit from a larger architecture, and we thus include both variants for a fair comparison. We experiment with cross-task transfer on three subsets of tasks: tasks that share *similar* game mechanics (for which we consider two ***Shooter*** and ***Maze*** categories), and tasks that have no discernible properties in common (referred to as ***Diverse***). We measure performance on individual Atari games by absolute scores, and also provide aggregate results as measured by mean and median scores across games, normalized by human performance or EfficientZero performance at 100k environment steps. All of our results are averaged across 5 random seeds (see Appendix D for more details). We provide details on our pretraining dataset in Appendix F.

**Baselines.** We compare our method against **7** prior methods for online RL that represent the state-of-the-art on the Atari100k benchmark (including EfficientZero), a multi-task behavior cloning policy pretrained on the same offline data as our method does for zero-shot performance on the target task and the performance after finetuning on the target task (see Appendix O for details), and a direct comparison to CURL (Srinivas et al., 2020), a strong model-free RL baseline, under an offline pretraining + online finetuning setting. We also include a set of ablations that include EfficientZero with several different model sizes and pretraining/finetuning schemes. The former baselines serve to position our results with respect to the state-of-the-art, and the latter baselines and ablations serve to shed light on the key ingredients for successful multi-task pretraining and finetuning.

### 4.1 RESULTS & DISCUSSION

We introduce our results in the context of each of our three questions, and discuss our main findings.

*1. How does our proposed framework compare to alternative pretraining and online RL approaches with limited online interaction from the target task?*

**Tasks with *similar* game mechanics.** We first investigate the feasibility of finetuning models that are pretrained on games with *similar* mechanics. We select 5 shooter games and 5 maze games for this experiment. Results for our method, baselines, and a set of ablations on the Atari100k benchmark

*Table 2.* **Atari100k benchmark results (diverse pretraining tasks).** XTRA results use the *same* set of pretrained model parameters obtained by offline pretraining on 8 diverse games. Mean of 5 seeds each with 32 evaluation episodes. Our result is highlighted . All other results are adopted from EfficientZero (Ye et al., 2021). We also report human-normalized mean and median scores.

| Game | XTRA (Ours) | EfficientZero | Random | Human | SimPLe | OTRainbow | DrQ | SPR | MuZero | CURL |
|---|---|---|---|---|---|---|---|---|---|---|
| Assault | **1742.2** | 1263.1 | 222.4 | 742.0 | 527.2 | 351.9 | 452.4 | 571.0 | 500.1 | 600.6 |
| BattleZone | 14631.3 | 13871.2 | 2360.0 | 37187.5 | 5184.4 | 4060.6 | 12954.0 | **16651.0** | 7687.5 | 14870.0 |
| Hero | **10631.8** | 9315.9 | 1027.0 | 30826.4 | 2656.6 | 6458.8 | 3736.3 | 7019.2 | 3095.0 | 6279.3 |
| Krull | **7735.8** | 5663.3 | 1598.0 | 2665.5 | 4539.9 | 3277.9 | 4018.1 | 3688.9 | 4890.8 | 4229.6 |
| Seaquest | 749.5 | 1100.2 | 68.4 | 42054.7 | 683.3 | 286.9 | 301.2 | 583.1 | 208.0 | 384.5 |
| Normed Mean | **1.87** | 1.29 | 0.00 | 1.00 | 0.70 | 0.41 | 0.62 | 0.65 | 0.77 | 0.75 |
| Normed Median | 0.35 | 0.33 | 0.00 | 1.00 | 0.08 | 0.18 | 0.30 | **0.41** | 0.15 | 0.36 |

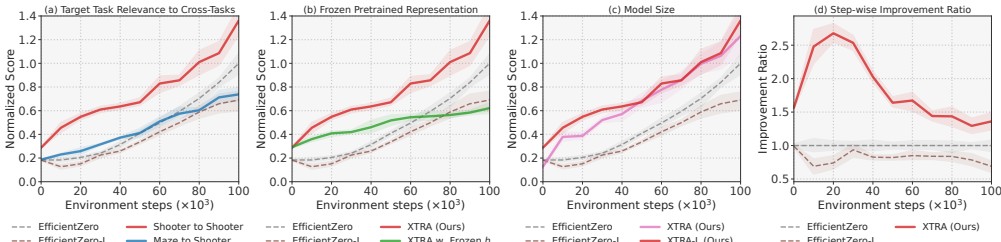

*Figure 6.* **(a)** Effectiveness of Task Relevance, **(b)** Frozen Representation, **(c)** Model Size, and **(d)** Environment Steps. We visualize model performance on aggregated scores (5 seeds) from 5 shooter games.

are shown in Table 1. For completeness, we also provide learning curves in Figure 11 as well as an aggregate curve across 5 seeds of all 10 games normalized by EfficientZero's performance at 100K environment steps in Figure 1. We find that pretraining improves sample-efficiency substantially across most tasks, improving mean and median performance of EfficientZero by **23**% and **25**%, respectively, overall. Interestingly, XTRA also had a notable zero-shot ability compared to a multi-game Behavioral Cloning baseline that is trained on the same offline dataset. We also consider three ablations: *(1)* **XTRA without cross-task:** a variant of our method that naively finetunes the pretrained model without any additional offline data from pretraining tasks during finetuning, *(2)* **XTRA without pretraining:** a variant that uses our concurrent cross-task learning (*i.e., leverages offline data during finetuning*) but is initialized with random parameters (no pretraining), and finally *(3)* **XTRA without task weights:** a variant that uses constant weights of 1 for all task loss terms during finetuning. We find that XTRA achieves extremely high performance on 2 games (DemonAttack and Phoenix) without dynamic task weights, improving over EfficientZero by as much as **343**% on DemonAttack. However, its median performance is overall low compared to our default variant that uses dynamic weights. We conjecture that this is because some (combinations of) games are more susceptible to gradient conflicts than others.

**Tasks with *diverse* game mechanics.** We now consider a more diverse set of pretraining and target games that have no discernible properties in common. Specifically, we use the following tasks for pretraining: Carnival, Centipede, Phoenix, Pooyan, Riverraid, VideoPinball, WizardOfWor, and YarsRevenge, and evaluate our method on 5 tasks from Atari100k. Results are shown in Table 2. We find that XTRA advances the state-of-the-art in a majority of tasks on the Atari100k benchmark, and achieve a mean human-normalized score of **187**% vs. **129**% for the previous SOTA, EfficientZero. We perform the same set of the ablations as we do for tasks with similar game mechanics with XTRA, and the results are shown in Table 6 from Appendix H. Additionally, we include an ablation that examines the effect of the number of pretrained tasks on later finetuning performance. Details and results for this ablation are shown in Table 7 from Appendix I.

**Model-free comparisons.** For both settings (e.g., tasks with similar & diverse game mechanics), we also compare our framework with a strong model-free baseline, CURL (Srinivas et al., 2020), where CURL is pretrained on the same pretraining tasks as XTRA is, and later finetuned to each of the target tasks. We find that pretraining does not improve the performance of this model-free baseline as consistently as for our model-based framework, XTRA, under both settings. More details and results on this comparison can be found in Table 4 and 5 from Appendix G.

### 2. *How do the individual components of our framework influence its success?*

**A deeper look at task relevance.** While our previous experiments established that XTRA benefits from pretraining even when games are markedly different, we now seek to better quantify the

importance of task relevance. We compare the online finetuning performance of XTRA in two different settings: *(1)* pretraining on 4 *shooter* games and finetuning to 5 new *shooter* games, and *(2)* pretraining on 4 *maze* games and finetuning to the same 5 *shooter* games. Aggregate results across all 5 target tasks are shown in Figure 6 *(a)*. Unsurprisingly, we observe that offline pretraining and concurrently learning from other shooter games significantly benefit the online target shooter games through training, with particularly large gains early in training. On the contrary, pretraining on maze games and finetuning to shooter games show similar performance compared to EfficientZero trained from scratch. This result indicates that *(1)* selecting pretraining tasks that are relevant to the target task is key to benefit from pretraining, and *(2)* in the extreme case where there are *no* pretraining tasks relevant to the target task, finetuning with XTRA generally does not harm the online RL performance since it can automatically assign small weights to the pretraining tasks.

**Which components transfer in model-based RL?** Next, we investigate which model component(s) are important to the success of cross-task transfer. We answer this question by only transferring a subset of the different model components – representation $h$, dynamics function $g$, and prediction head $f$ – to the online finetuning stage and simply using a random initialization for the remaining components. Results are shown in Figure 7. Interestingly, we find that only transferring the pretrained representation $h$ to the online RL stage only improves slightly over learning from scratch, especially in the early stages of online RL. In comparison, loading both the pretrained representation *and* dynamics function accounts for the majority of the gains in XTRA, whereas loading the prediction heads has no significant impact on sample-efficiency (but matters for zero-shot performance). We conjecture that this is because learning a good dynamics function is relatively more difficult from few samples than learning a *task-specific* visual representation, and that the prediction head accounts for only a small amount of the overall parameters in the model. Finally, we hypothesize that the visual representation learned during pretraining will be susceptible to distribution shifts as it is transferred to an unseen target task. To verify this hypothesis, we consider an additional experiment where we transfer all components to new tasks, but *freeze* the representation $h$ during finetuning, *i.e.*, it remains fixed. Results for this experiment are shown in Figure 6 *(b)*. We find that, although this variant of our framework improves over training from scratch in the early stages of training, the frozen representation eventually hinders the model from converging to a good model, which is consistent with observations made in (supervised) imitation learning (Parisi et al., 2022).

**Scaling model size.** In this experiment, we investigate whether XTRA benefits from larger model sizes. Since dynamics network $g$ and prediction network $f$ are used in MCTS search, increasing the parameter counts for these two networks would increase inference/training time complexity significantly. However, increasing the size of the representation network $h$ has a relatively small impact on overall inference/training time (see Figure 3 for reference). We compare the performance of our method and EfficientZero trained from scratch with each of our two model sizes, the original EfficientZero architecture and a larger variant (denoted EfficientZero-**L**); results are shown in Figure 6 *(c)*. We find that our default, larger variant of XTRA (denoted XTRA-L in the figure) is slightly better than the smaller model size. In comparison, EfficientZero-L, performs significantly worse than the smaller variant of EfficientZero.

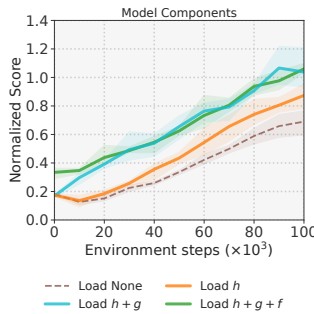

*Figure 7.* **Effectiveness of model components.** The aggregated scores from 5 shooter games by loading parameters of different pretrained model components. Mean of 5 seeds; shaded area indicates 95% CIs.

**Relative improvement vs. environment steps.** Finally, we visualize the average improvement over EfficientZero throughout training in Figure 6 *(d)*. Results show that XTRA is particularly useful in the early stages of training, *i.e.*, in an extremely limited data setting. We therefore envision that cross-task pretraining could benefit many real-world applications of RL, where environment interactions are typically constrained due to physical constraints.

### 3. When can we empirically expect finetuning to be successful?

Based on Table 1 and 2, we conclude that cross-task transfer with model-based RL is feasible. Further, Figure 6 *(a)* shows that our XTRA framework benefits from online finetuning when pretraining tasks are relevant, and both representation and dynamics networks contribute to its success (Figure 7).

## 5 RELATED WORK

**Pretrained representations** are widely used to improve downstream performance in learning tasks with limited data or resources available, and have been adopted across a multitude of areas such as computer vision (Girshick et al., 2014; Doersch et al., 2015; He et al., 2020), natural language processing (Devlin et al., 2019; Brown et al., 2020), and audio (van den Oord et al., 2018). By first learning a good representation on a large dataset, the representation can quickly be finetuned with, *e.g.*, supervised learning on a small labelled dataset to solve a given task (Pan & Yang, 2010). For example, He et al. (2020) show that contrastive pretraining on a large, unlabelled dataset learns good features for ImageNet classification, and Brown et al. (2020) show that a generative model trained on large-scale natural language data can be used to solve unseen tasks given only a few examples. While this is a common paradigm for problems that can be cast as (self-)supervised learning problems, it has seen comparably little adoption in RL literature. This discrepancy is, in part, attributed to optimization challenges in RL (Bodnar et al., 2020; Hansen et al., 2021b; Xiao et al., 2022; Wang et al., 2022), as well as a lack of large-scale datasets that capture both the visual, temporal, and control-relevant (actions, rewards) properties of RL (Hansen et al., 2022b). In this work, we show that – despite these challenges – modern model-based RL algorithms can still benefit substantially from pretraining on multi-task datasets, but require a more careful treatment during finetuning.

**Sample-efficient RL.** Improving the sample-efficiency of visual RL algorithms is a long-standing problem and has been approached from many – largely orthogonal – perspectives, including representation learning (Kulkarni et al., 2019; Yarats et al., 2019; Srinivas et al., 2020; Schwarzer et al., 2021), data augmentation (Laskin et al., 2020; Kostrikov et al., 2021; Hansen et al., 2021b), bootstrapping from demonstrations (Zhan et al., 2020) or offline datasets (Wang et al., 2022; Zheng et al., 2022; Baker et al., 2022), using pretrained visual representations for model-free RL (Shah & Kumar, 2021; Xiao et al., 2022; Ze et al., 2022), and model-based RL (Ha & Schmidhuber, 2018; Finn & Levine, 2017; Nair et al., 2018; Hafner et al., 2019b; Kaiser et al., 2020; Schrittwieser et al., 2020; Hafner et al., 2021; Ye et al., 2021; Hansen et al., 2022a; Seo et al., 2022; Hansen et al., 2023). We choose to focus our efforts on sample-efficiency from the perspective of pretraining in a model-based context, *i.e.*, jointly learning perception *and* dynamics. Several prior works consider these problems independently from each other: Xiao et al. (2022) shows that model-free policies can be trained with a frozen pretrained visual backbone, and Seo et al. (2022) shows that learning a world model on top of features from a visual backbone pretrained with video prediction can improve model learning. Our work differs from prior work in that we show it is possible to pretrain *and* finetune both the representation *and* the dynamics using model-based RL.

**Finetuning in RL.** Gradient-based finetuning is a well-studied technique for adaptation in (predominantly model-free) RL, and has been used to adapt to either changes in visuals or dynamics (Mishra et al., 2017; Yen-Chen et al., 2019; Duan et al., 2016; Julian et al., 2020; Hansen et al., 2021a; Bodnar et al., 2020; Wang et al., 2022; Ze et al., 2022), or task specification (Xie & Finn, 2021; Walke et al., 2022). For example, Julian et al. (2020) shows that a model-free policy for robotic manipulation can adapt to changes in lighting and object shape by finetuning via rewards on a mixture of data from the new and old environment, and recover original performance in less than 800 trials. Similarly, Hansen et al. (2021a) shows that model-free policies can (to some extent) also adapt to small domain shifts by self-supervised finetuning within a single episode. Other works show that pretraining with offline RL on a dataset from a specific task improve sample-efficiency during online finetuning on the same task (Zheng et al., 2022; Wang et al., 2022). Finally, Lee et al. (2022) shows that offline multi-task RL pretraining via sequence modelling can improve offline finetuning on data from unseen tasks. Our approach is most similar to Julian et al. (2020) in that we finetune via rewards on a mixture of datasets. However, our problem setting is fundamentally different: we investigate whether *multi-task* pretraining can improve online RL on an *unseen* task across *multiple* axes of variation.

## 6 CONCLUSION

In this paper, we propose Model-Based **Cross**-Task **Tra**nsfer (**XTRA**), a framework for sample-efficient online RL with scalable pretraining and finetuning of learned world models using extra, auxiliary data from other tasks. We find that XTRA improves sample-efficiency substantially across most tasks, improving mean and median performance of EfficientZero by 23% and 25%, respectively, overall. As a feasibility study, we hope that our empirical analysis and findings on cross-task transfer with model-based RL will inspire further research in this direction.

**Reproducibility Statement.** We base all experiments on the publicly available Atari100k benchmark, and provide extensive implementation details in the appendices, including detailed network architecture and hyper-parameters. Code for reproducing our experiments is made available at https://nicklashansen.github.io/xtra.

**Acknowledgements.** This work is supported by NSF Award IIS-2127544, NSF TILOS AI Institute, and gifts from Qualcomm AI. We would like to thank Weirui Ye, Wenlong Huang, and Weijian Xu for helpful discussions.

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

## A  XTRA/EFFICIENTZERO OBJECTIVES

XTRA uses the same learning objective as EfficientZero during both offline pretraining and online finetuning, except the quantity targets are predicted by the teacher model during distillation and concurrent learning for pretrained games instead of Muzero Reanalysis procedure.

Here, we explain the objectives of EfficientZero (Ye et al., 2021) and its predecessor MuZero (Schrittwieser et al., 2020). To warrant the latent dynamics that can mirror the true states of the environment, MuZero is trained to predict three necessary quantities directly relevant for planning: (1) the policy target $\pi$ obtained from visit count distribution of the MCTS (2) immediate reward $u$ from environment (3) bootstrapped value target $z$ where $z = \sum_{i=0}^{k-1} \gamma^i u^i + \gamma^k v_{t+k}$. On top of MuZero, EfficientZero adds a self-supervised consistency loss term, and predicts sum of environment rewards from next $k$ steps, $\sum_{i=0}^{k-1} \gamma^i u^i$, instead of single-step reward. We refer reader to the original manuscripts for implementation details. We present the learning objective for EfficientZero at time step $t$ with $k$ unroll steps:

$$\mathcal{L}_t^{ez}(\theta) = \sum_{k=0}^{K} \underbrace{\|\mathcal{L}^r(u_{t+k}, \hat{r}_t^k)\|_2^2}_{\text{reward}} + \lambda_1 \underbrace{\|\mathcal{L}^p(\pi_{t+k}, \hat{p}_t^k)\|_2^2}_{\text{policy}} + \lambda_2 \underbrace{\|\mathcal{L}^v(z_{t+k}, \hat{v}_t^k)\|_2^2}_{\text{value}} + \lambda_3 \underbrace{\|\mathcal{L}^s(s_{t+1}, \hat{s}_{t+1})\|_2^2}_{\text{consistency}} + c\|\theta\| \tag{3}$$

## B  TASK WEIGHTS COMPUTATION

Within the $N$-steps update, we maintain a task-specific counter $s^i$ and update the counter by $\Delta s_n^i$ at each step $n$ as follows:

$$\Delta s_n^i = \begin{cases} 1, & \text{if}\quad \text{Sim}(\tilde{\mathcal{M}}^i, \mathcal{M}) > 0.1 \\ 0, & \text{otherwise} \end{cases}$$
$$s^i = s^i + \Delta s_n^i \tag{4}$$

At every $N$ steps, the new task weights $\eta^i$ are updated by $\eta^i = \frac{s^i}{N}$, and used in subsequent finetuning objectives according to Equation 1. In practice, we start task weight updates at 10k steps to ensure enough data from the online target task has been collected for a meaningful similarity measure. All task weights are initialized as 1 for the first 10k steps.

Figure 8 shows how task weights are adaptively adjusted by the model during 100k environment steps during online finetuning stage for 10 games reported in Figure 11. Figure 9 shows the adjustments of task weights for 5 games reported in Figure 12.

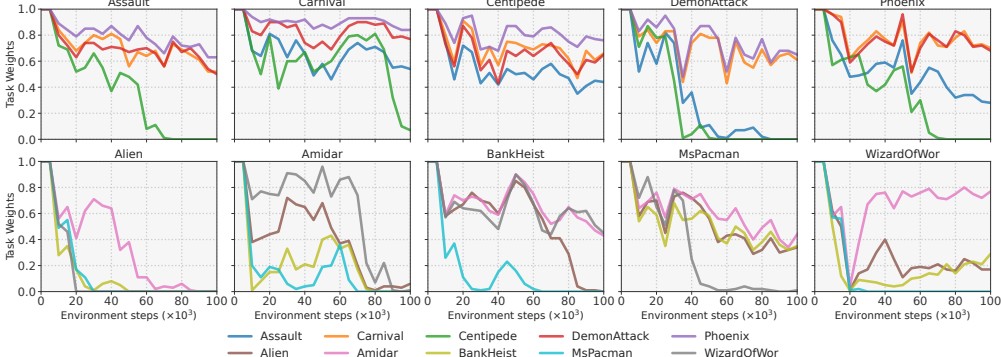

*Figure 8.* **Periodic task re-weighting.** We visualize task weights as a function of environment steps for each of our 10 tasks from Table 1. First row corresponds to *shooter* games and the bottom row corresponds to *maze* games. We evaluate task weights on all tasks from the same category except for the target task itself.

## C  DISTILLATION VS. MULTI-GAME OFFLINE RL

Our method learns a multi-game world model from offline data via distillation of task-specific world models trained with offline RL. An alternative way to obtain such a multi-game model would be

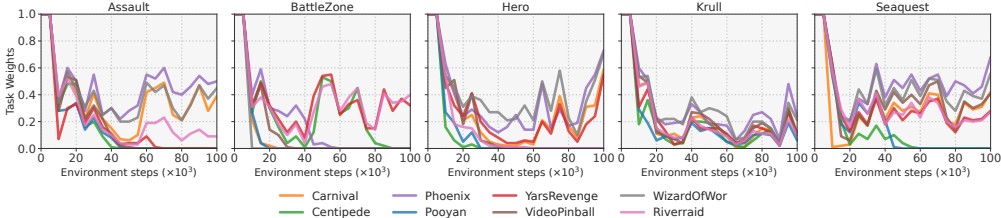

*Figure 9.* **Periodic task re-weighting.** We visualize task weights as a function of environment steps for each of our 5 tasks from Table 2. We evaluate task weights on all 8 tasks used during pretraining.

to directly train a single world model on the multi-game dataset with offline RL. However, we find that learning such a model is difficult. A comparison between the two approaches on four different pretraining games is shown in Figure 10. We observe that multi-game offline RL (*green*) achieves low scores in all four pretraining games, whereas our wold model obtained by distillation (*orange*) performs comparably to single-task world models (*blue*) in 3 out of 4 tasks.

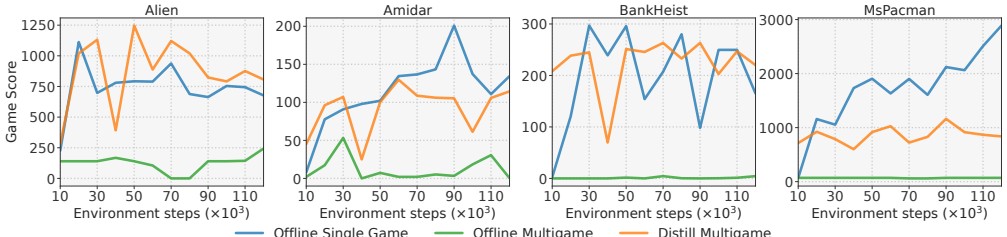

*Figure 10.* **Distillation vs. multi-game offline RL.** Results are shown on four pretraining tasks. Single-game models *(blue)* are trained via offline RL on each individual task, whereas results for multi-game offline RL (*green*) and our proposed distillation (*orange*) are obtained by evaluating a single set of pretraining parameters. Distillation nearly matches single-task performance.

# D   SCORES FOR INDIVIDUAL SEEDS

Our results in Table 1 are aggregated across 5 seeds. In Table 3, we report game scores for each individual seed, as well as the mean, median, and standard deviation of game scores for each game. We also list random and human scores obtained from Badia et al. (2020), and calculate the Human Normalized Score based on the formula: $(\text{score}_{\text{agent}} - \text{score}_{\text{random}})/(\text{score}_{\text{human}} - \text{score}_{\text{random}})$ as in prior work. To the best of our knowledge, there are no human performance results for the Carnival game, and we therefore exclude this game from the aggregate Human Normalized Mean and Median Scores computed in Table 3.

*Table 3.* **Scores for individual seeds.** Per-seed game scores for our method, a random behavior baseline, and human performance. We report both unnormalized and normalized scores (Human Normalized Scores), as well as their aggregate results. Random and human scores are obtained from Badia et al. (2020). We evaluate each random seed on 32 evaluation episodes at 100k steps.

| Game | Game Score per Seed | | | | | Aggregated Metrics | | | References | | Human Normed |
|---|---|---|---|---|---|---|---|---|---|---|---|
| | Seed 0 | Seed 1 | Seed 2 | Seed 3 | Seed 4 | Mean | Median | Std | Random | Human | |
| Assault | 1450.19 | 1356.53 | 1236.19 | 1215.12 | 1214.72 | 1294.55 | 1236.19 | 105.06 | 222.40 | 742.00 | 2.06 |
| Carnival | 3865.31 | 4867.50 | 4155.62 | 2601.88 | 3814.38 | 3860.94 | 3865.31 | 819.67 | - | - | |
| Centipede | 7596.38 | 6179.25 | 5380.41 | 5300.03 | 3950.88 | 5681.39 | 5380.41 | 1336.58 | 2090.90 | 12017.00 | 0.36 |
| DemonAttack | 10470.78 | 8051.25 | 27574.06 | 8117.81 | 16490.47 | 14140.88 | 10470.78 | 8258.35 | 152.10 | 1971.00 | 7.69 |
| Phoenix | 20875.94 | 10988.44 | 10521.88 | 15803.44 | 14709.06 | 14579.75 | 14709.06 | 4198.81 | 761.40 | 7242.60 | 2.13 |
| Alien | 569.69 | 807.50 | 814.06 | 1388.12 | 1194.38 | 954.75 | 814.06 | 329.77 | 227.80 | 7127.70 | 0.11 |
| Amidar | 93.00 | 104.34 | 76.47 | 97.38 | 79.59 | 90.16 | 93.00 | 11.84 | 5.80 | 1719.50 | 0.05 |
| BankHeist | 303.12 | 316.56 | 270.62 | 270.00 | 364.06 | 304.88 | 303.12 | 38.83 | 14.20 | 753.10 | 0.39 |
| MsPacman | 1109.69 | 1960.00 | 1865.94 | 1228.44 | 1134.38 | 1459.69 | 1228.44 | 417.28 | 307.30 | 6951.60 | 0.17 |
| WizardOfWor | 1275.00 | 687.50 | 1056.25 | 990.62 | 915.62 | 985.00 | 990.62 | 213.62 | 563.50 | 4756.50 | 0.10 |
| Human Normed Mean | | | | | | | | | | | 1.45 |
| Human Normed Median | | | | | | | | | | | 0.36 |

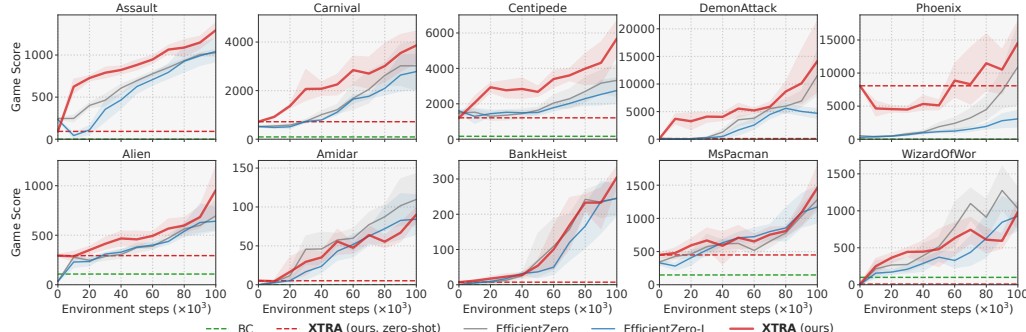

*Figure 11.* **Atari100k benchmark results (*similar* pretraining tasks).** We report unnormalized scores, aggregated across 5 seeds per game. Shaded area indicates 95% confidence intervals.

# E    ADDITIONAL EVALUATION CURVES OF XTRA ON ATARI100K BENCHMARK

For completeness, Figure 11 and 12 include evaluation curves of XTRA on the games for which we report final performance in Table 1 and 2.

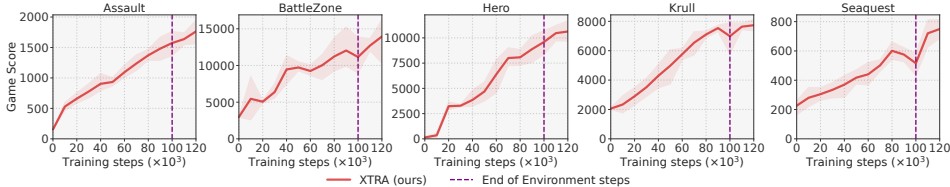

*Figure 12.* **Atari100k benchmark results (*diverse* pretraining tasks).** We report unnormalized scores, aggregated across 5 seeds per game. Shaded area indicates 95% confidence intervals.

# F    OFFLINE DATA PREPARATION

To train the model in offline multi-task pretraining stage, we use trajectories collected by EfficientZero (Ye et al., 2021) on the Atari100k benchmark. For each pretraining game, we assume we can access model checkpoints obtained every 10k steps from 120k training steps (the environment step is capped at 100k), resulting in 12 model checkpoints. For each checkpoint, we evaluate the model performance on the Atari environment following the same procedure from EfficientZero and collect 64 trajectories. This translates to an average of 1M transitions per game, but varies depending on episode length – for example, this only results in 636k transitions for the game of Assault. Since trajectories are collected from model checkpoints both at the early and late training stage within the 120k training steps, the collected data does not necessarily come from an expert agent. Thus, we show that XTRA is effective even when pretraining data is suboptimal, allowing us to learn from very diverse data sources.

# G    PRETRAINING + FINETUNING IN MODEL-FREE RL

We compare the effectiveness of our framework, XTRA, with a strong model-free baseline, CURL, that we also implement following a similar pretraining and finetuning scheme. This is in contrast to the original formulation of CURL that does not leverage pretraining. We implement our training scheme for CURL with the following setup: *(1)* pretrain a multi-task CURL model on the same pretraining tasks as our framework uses (using offline data generated from training individual CURL agents), and *(2)* directly finetune the pretrained model on the target task with online RL for 100k environment steps.

*Table 4.* **Model-based XTRA comparison with the model-free pretraining + finetuning scheme for tasks that share *diverse* game mechanics.** The model is pretrained with Carnival, Centipede, Phoenix, Pooyan, Riverraid, VideoPinball, WizardOfWor, and YarsRevenge. Results for EfficientZero, CURL, Random, and Human are adopted from EfficientZero (Ye et al., 2021). All other results are based on the average of 5 runs.

| Game | Model-Based | | Model-Free | | | |
|---|---|---|---|---|---|---|
| | Efficient Zero | XTRA (Ours) | CURL | CURL$^{ft}$ | Random | Human |
| Assault | 1263.1 | **1742.2** | 600.6 | 588.6 | 222.4 | 742.0 |
| BattleZone | 13871.2 | 14631.3 | 14870.0 | **16450.0** | 2360.0 | 37187.5 |
| Hero | 9315.9 | **10631.8** | 6279.3 | 6294.5 | 1027.0 | 30826.4 |
| Krull | 5663.3 | **7735.8** | 4229.6 | 3472.8 | 1598.0 | 2665.5 |
| Seaquest | **1100.2** | 749.5 | 384.5 | 385.5 | 68.4 | 42054.7 |
| Normed Mean | 1.29 | **1.87** | 0.75 | 0.60 | 0.00 | 1.00 |
| Normed Median | 0.33 | **0.35** | 0.36 | 0.40 | 0.00 | 1.00 |

*Table 5.* **Model-based XTRA comparison with the model-free pretraining + finetuning scheme for tasks that share *similar* game mechanics**. For each target game for finetuning, the model is first pretrained on all other 4 games from the same category. All results are based on the average of 5 runs.

| Category | Game | Model-Based | | Model-Free | |
|---|---|---|---|---|---|
| | | Efficient Zero | XTRA (Ours) | CURL | CURL$^{ft}$ |
| ***Shooter*** | Assault | 1027.1 | **1294.6** | 590.2 | 461.2 |
| | Carnival | 3022.1 | **3860.9** | 591.6 | 714.8 |
| | Centipede | 3322.7 | **5681.4** | 4137.7 | 3731.0 |
| | DemonAttack | 11523.0 | **14140.9** | 908.3 | 638.9 |
| | Phoenix | 10954.9 | **14579.8** | 901.2 | 1168.4 |
| | Mean Improvement | 1.00 | **1.36** | 0.44 | 0.39 |
| | Median Improvement | 1.00 | **1.28** | 0.20 | 0.24 |
| ***Maze*** | Alien | 695.0 | **954.8** | 905.2 | 782.6 |
| | Amidar | 109.7 | 90.2 | 109.7 | **169.9** |
| | BankHeist | 246.1 | **304.9** | 151.8 | 86.2 |
| | MsPacman | 1281.4 | **1459.7** | 1421.6 | 1234.1 |
| | WizardOfWor | 1033.1 | 985 | **1262.0** | 1244.4 |
| | Mean Improvement | 1.00 | **1.11** | 1.05 | 1.04 |
| | Median Improvement | 1.00 | **1.14** | 1.11 | 1.13 |
| ***Overall*** | Mean Improvement | 1.00 | **1.23** | 0.74 | 0.72 |
| | Median Improvement | 1.00 | **1.25** | 0.81 | 0.71 |

# H    XTRA ABLATIONS FOR TASKS WITH DIVERSE GAME MECHANICS

We perform the same set of ablations for tasks that share *diverse* game mechanics as we do for tasks that share *similar* game mechanics for XTRA. Results are shown in Table 6. Details of each ablation can be found in Section 4.1.

*Table 6.* **XTRA ablation for tasks that share *diverse* game mechanics**. Results for EfficientZero, Random, and Human are adopted from EfficientZero (Ye et al., 2021). All other results are based on the average of 5 runs.

| Game | | | Ablations (XTRA) | | | | |
|---|---|---|---|---|---|---|---|
| | Efficient Zero | XTRA (Ours) | w.o. cross-task | w.o. pretraining | w.o. task weights | Random | Human |
| Assault | 1263.1 | **1742.2** | 1716.11 | 1183.58 | 1605.07 | 222.4 | 742.0 |
| BattleZone | 13871.2 | **14631.3** | 12918.8 | 8718.8 | 10087.5 | 2360.0 | 37187.5 |
| Hero | 9315.9 | **10631.8** | 8275.3 | 8672.9 | 7755.4 | 1027.0 | 30826.4 |
| Krull | 5663.3 | **7735.8** | 5910.7 | 6767.3 | 7104.7 | 1598.0 | 2665.5 |
| Seaquest | 1100.2 | 749.5 | **811.4** | 540.6 | 493.1 | 68.4 | 42054.7 |
| Normed Mean | 1.29 | **1.87** | 1.50 | 1.43 | 1.66 | 0.00 | 1.00 |
| Normed Median | 0.33 | **0.35** | 0.30 | 0.26 | 0.23 | 0.00 | 1.00 |

*Table 7*. **XTRA ablation (number of tasks in pretraining & cross-task finetuning) for tasks that share** *diverse* **game mechanics**. Results for EfficientZero are adopted from EfficientZero (Ye et al., 2021). All other results are based on the average of 5 runs.

| Game | XTRA | Ablations (XTRA) | | | EfficientZero |
|------|------|---------|---------|---------|---------------|
| | **8 Games** | **4 Games** | **2 Games** | **0 Games** | **0 Games** |
| Assault | **1742.2** | 1676.7 | 1463.8 | 1255.9 | 1263.1 |
| BattleZone | **14631.3** | 9581.3 | 9550.0 | 10125.0 | 13871.2 |
| Hero | **10631.8** | 9654.9 | 8506.5 | 6815.1 | 9315.9 |
| Krull | **7735.8** | 7375.6 | 7348.9 | 5590.6 | 5663.3 |
| Seaquest | 749.5 | 656.4 | 627.5 | 770.8 | **1100.2** |
| Normed Mean | **1.87** | 1.74 | 1.65 | 1.23 | 1.29 |
| Normed Median | **0.35** | 0.29 | 0.25 | 0.22 | 0.33 |

## I    Effects of Number of Tasks in Pretraining and Cross-Tasks in Finetuning

We perform an additional ablation for tasks that share *diverse* game mechanics – whether changing the number of tasks during pretraining would help or hurt the performance in later cross-task finetuning. By reducing the number of tasks in pretraining, the model is exposed to (1) less diverse game mechanics and (2) less offline training data in pretraining, and fewer cross-tasks in finetuning. In this ablation, we gradually reduce the number of pretrained tasks from 8 (Carnival, Centipede, Phoenix, Pooyan, Riverraid, VideoPinball, WizardOfWor, and YarsRevenge), to 4 (Phoenix, WizardOfWor, VideoPinball, YarsRevenge), and to 2 (Phoenix, VideoPinball). XTRA is reduced to EfficientZero-L when the number of pretrained tasks and cross-tasks during finetuning is set to 0. We find that increasing the number of tasks during pretraining (and later cross-task finetuning) mostly consistently improves XTRA performance.

## J    Architectural Details

We adopt the architecture of EfficientZero (Ye et al., 2021). For EfficientZero-L and XTRA, we increase the number of residual blocks from 1 (default) to 4 (ours) in the representation network, which we find to improve pretraining slightly for XTRA. However, we find that our baseline EfficientZero (without pretraining) performs significantly worse with a larger representation network. Therefore, we use the default EfficientZero as the main point of comparison throughout this work and only include Efficient-L for completeness.

The architecture of the **representation networks** is as follows:
- 1 convolution with stride 2 and 32 output planes, output resolution 48x48. (BN + ReLU)
- 1 residual block with 32 planes.
- 1 residual downsample block with stride 2 and 64 output planes, output resolution 24x24.
- 1 residual block with 64 planes.
- Average pooling with stride 2, output resolution 12x12. (BN + ReLU)
- 1 residual block with 64 planes.
- Average pooling with stride 2, output resolution 6x6. (BN + ReLU)
- 1 residual block with 64 planes.

, where the kernel size is $3 \times 3$ for all operations.

The architecture of the **dynamics networks** is as follows:
- Concatenate the input states and input actions into 65 planes.
- 1 convolution with stride 2 and 64 output planes. (BN)
- A residual link: add up the output and the input states. (ReLU)
- 1 residual block with 64 planes.

The architecture of the **reward prediction network** is as follows:
- 1 1x1convolution and 16 output planes. (BN + ReLU)
- Flatten.

- LSTM with 512 hidden size. (BN + ReLU)
- 1 fully connected layers and 32 output dimensions. (BN + ReLU)
- 1 fully connected layers and 601 output dimensions.

The architecture of the **value and policy prediction networks** is as follows:

- 1 residual block with 64 planes.
- 1 1x1convolution and 16 output planes. (BN + ReLU)
- Flatten.
- 1 fully connected layers and 32 output dimensions. (BN + ReLU)
- 1 fully connected layers and $D$ output dimensions.

where $D = 601$ in the value prediction network and $D = |\mathcal{A}|$ in the policy prediction network.

## K    HYPER-PARAMETERS

We adopt our hyper-parameters from EfficientZero (Ye et al., 2021) with minimal modification. Because XTRA uses data from offline tasks to perform cross-task transfer during online finetuning, we have an additional hyper-parameter for mini-batch size for offline tasks, which is set to 256 (default). We list all hyper-parameters in Table 8 for completeness. Lastly, we note that EfficientZero performs an additional 20k gradient steps at 100k environment steps, with a $10\times$ smaller learning rate. We follow this procedure when comparing to previous state-of-the-art methods (Table 2), but for simplicity we omit these additional gradient steps in the remainder of our experiments for both XTRA and baselines.

*Table 8.* **Hyper-parameters.** We list all relevant hyper-parameters below. Values are adopted from Ye et al. (2021) with minimal modification but included here for completeness.

| Parameter | Setting |
|---|---|
| Observation down-sampling | $96 \times 96$ |
| Frames stacked | 4 |
| Frames skip | 4 |
| Reward clipping | True |
| Terminal on loss of life | True |
| Max frames per episode | 108K |
| Discount factor | $0.997^4$ |
| Minibatch size (offline tasks) | 256 |
| Minibatch size (target task) | 256 |
| Optimizer | SGD |
| Optimizer: learning rate | 0.2 |
| Optimizer: momentum | 0.9 |
| Optimizer: weight decay ($c$) | 0.0001 |
| Learning rate schedule | $0.2 \to 0.02$ |
| Max gradient norm | 5 |
| Priority exponent ($\alpha$) | 0.6 |
| Priority correction ($\beta$) | $0.4 \to 1$ |
| Training steps | 100K/120K |
| Evaluation episodes | 32 |
| Min replay size for sampling | 2000 |
| Self-play network updating inerval | 100 |
| Target network updating interval | 200 |
| Unroll steps ($l_{\text{unroll}}$) | 5 |
| TD steps ($k$) | 5 |
| Policy loss coefficient ($\lambda_1$) | 1 |
| Value loss coefficient ($\lambda_2$) | 0.25 |
| Self-supervised consistency loss coefficient ($\lambda_3$) | 2 |
| LSTM horizontal length ($\zeta$) | 5 |
| Dirichlet noise ratio ($\xi$) | 0.3 |
| Number of simulations in MCTS ($N_{\text{sim}}$) | 50 |
| Reanalyzed policy ratio | 1.0 |

## L  EFFECT OF MINI-BATCH SIZE

During online finetuning (stage 2), we finetune both with data from the target task, and data from the pretraining tasks. We maintain the same batch size (256) for the target task data as the non-pretraining baselines, but add additional data from the pretraining tasks with a 1:1 ratio. Thus, our effective batch size is $2\times$ that of the baselines. To verify that performance improvements stem from our pretraining (stage 1) and not the larger batch size, we compare our method to a variant of our EfficientZero-L baseline that uses a $2\times$ larger batch size (512). Results are shown in Figure 13. We do not observe any significant change in performance by doubling the batch size for the baseline. Thus, we conclude that a larger (effective) batch size is not the source of our performance gains, but rather our pretraining and inclusion of pretraining tasks during the online finetuning.

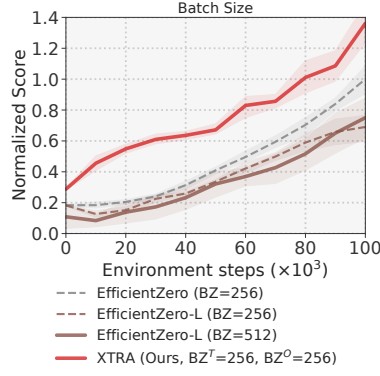

*Figure 13.* **Effect of mini-batch size (BZ).** For XTRA, we denote the batch size of the **T**arget task and **O**ffline tasks as $BZ^T$ and $BZ^O$, respectively.

## M  PER-GAME IMPROVEMENT OVER EFFICIENTZERO

We visualize XTRA's mean improvement over EfficientZero and EfficientZero-L under similar and diverse task settings on a per-game basis. The improvement is calculated based on the formula: $(\text{score}_{\text{XTRA}}/\text{score}_{\text{Baseline}}) - 1$. The Baseline can be either EfficientZero or EfficientZero-L, depending on the visualization setting.

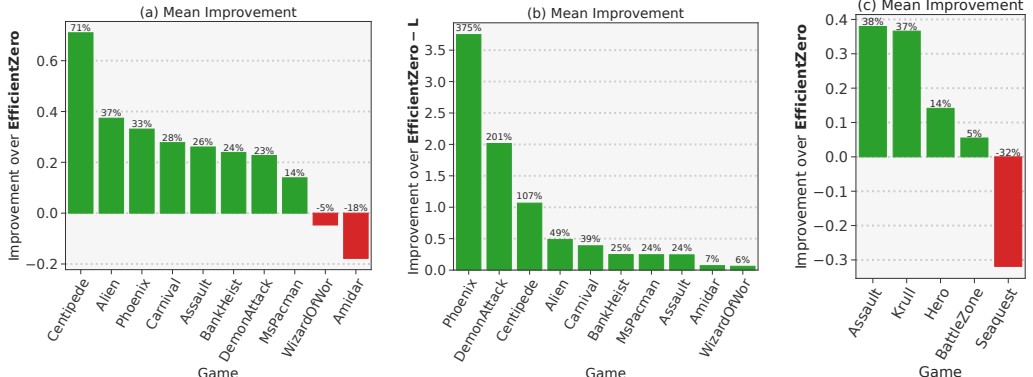

*Figure 14.* *(a)* Mean improvement over **EfficientZero** from Table 1 (*similar* games), *(b)* Mean improvement over **EfficientZero-L** from Table 1 (*similar* games), and *(c)* Mean improvement over **EfficientZero** from Table 2 (*diverse* games). Each result is aggregated across 5 seeds.

## N  GAME INFORMATION

In this section, we aim to provide additional context about the games that we consider during both pretraining and finetuning. Table 9 lists core properties for each game. The **Similar Task** column marks all games used in Table 1 (*similar* games), and the two **Diverse Task** columns mark all games used in Table 2 (*diverse* games) for pretraining and finetuning, respectively. We further categorize games into five categories based on game mechanics: *Maze*, *Shooter*, *Tank*, *Adventure*, and *Ball Tracking*, and also report whether the scene is static or dynamic, as well the (valid) action space for each task. The maximum dimensionality of the action space is 18 for Atari games.

## O  BEHAVIORAL CLONING BASELINE

We use the representation + prediction network in XTRA for the behavioral cloning (BC) study. The BC (finetune) from Table 1 follows an offline pretraining + offline finetuning paradigm. The model is finetuned on offline data for the target task (also generated by the EfficientZero baseline). We find

*Table 9.* **Game information.** We consider a variety of games in our experiments. Here, we provide more context to our selection of games. In our *similar* experiments (Table 1), we finetune model to each game after pretraining it on the other games within the same category. In *diverse* experiments (Table 2), we finetune a single model pretrained on all eight games to each of the target games.

| Games | Similar Task (Pretrain & Fine Tune) | Diverse Task (Pretrain) | Diverse Task (Fine Tune) | Category | Scene Continuity | Action Space |
|---|---|---|---|---|---|---|
| Alien | ✓ | | | Maze | | 18 |
| Amidar | ✓ | | | Maze | ✓ | 10 |
| Assault | ✓ | | ✓ | Shooter | ✓ | 7 |
| Bank Heist | ✓ | | | Maze | | 18 |
| Carnival | ✓ | ✓ | | Shooter | ✓ | 6 |
| Centipede | ✓ | ✓ | | Shooter | ✓ | 18 |
| DemonAttack | ✓ | | | Shooter | ✓ | 6 |
| MsPacman | ✓ | | | Maze | | 9 |
| Phoenix | ✓ | ✓ | | Shooter | | 8 |
| WizardOfWor | ✓ | ✓ | | Maze | | 10 |
| BattleZone | | | ✓ | Tank | ✓ | 18 |
| Hero | | | ✓ | Adventure | | 18 |
| Krull | | | ✓ | Adventure | | 18 |
| Seaquest | | | ✓ | Shooter | ✓ | 18 |
| Pooyan | | ✓ | | Shooter | | 6 |
| Riverraid | | ✓ | | Shooter | ✓ | 18 |
| VideoPinball | | ✓ | | Ball Tracking | | 9 |
| YarsRevenge | | ✓ | | Shooter | | 18 |

BC (finetune) underperforms the EfficientZero baseline. We also report zero-shot performance of the pretrained BC on their designated target tasks directly.

## P   GAME VISUALIZATIONS

We here provide sample trajectories for each game. These visualizations aim to highlight that *(1)* it is reasonable to expect some degree of cross-task transfer in *similar* game transfer (maze → maze, shooter → shooter) due to similarity in visuals and game mechanics, and *(2)* that games from our *diverse* category indeed are diverse, both in terms of visuals and mechanics. Figure 15 shows trajectories for five *Maze* games, Figure 16 shows trajectories for five *Shooter* games, and Figure 17 shows trajectories for eight games from our *diverse* game category.

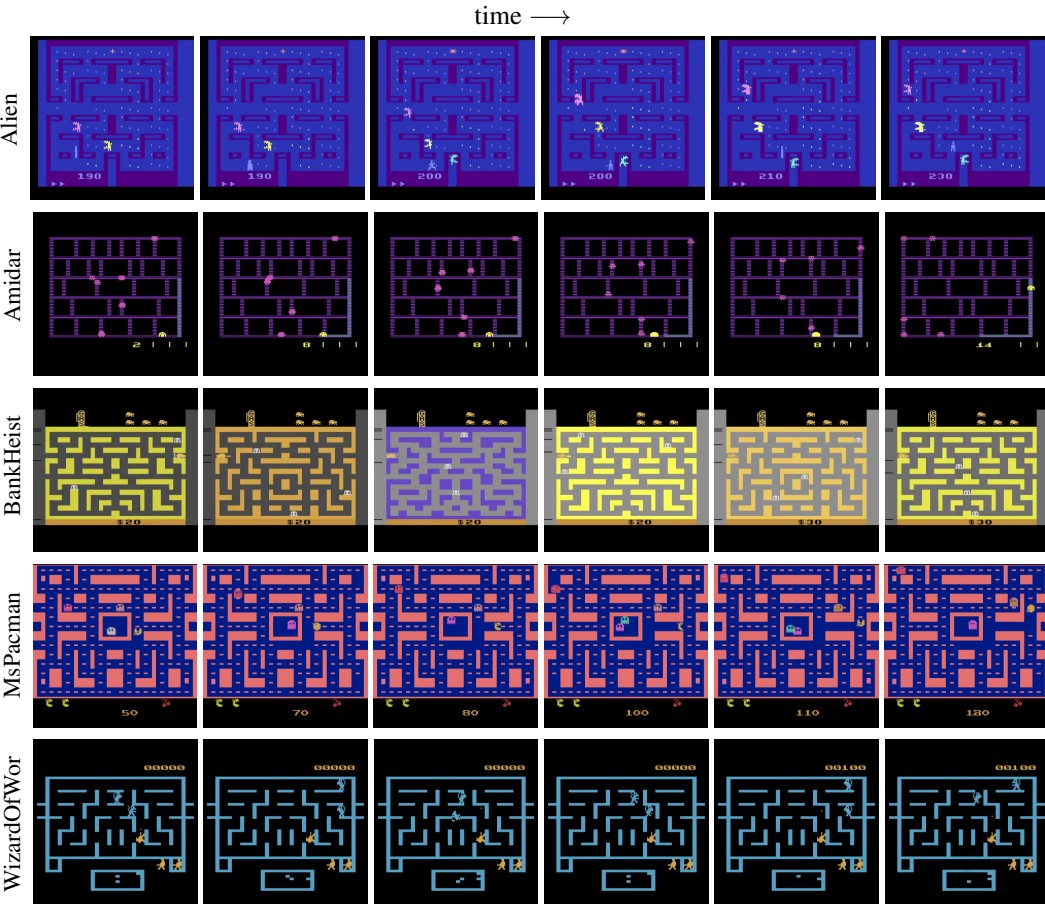

*Figure 15.* **Visualization of trajectories from the *Maze* game category.** We visualize key frames in sample trajectories for five tasks. Actual trajectory lengths vary greatly between games.

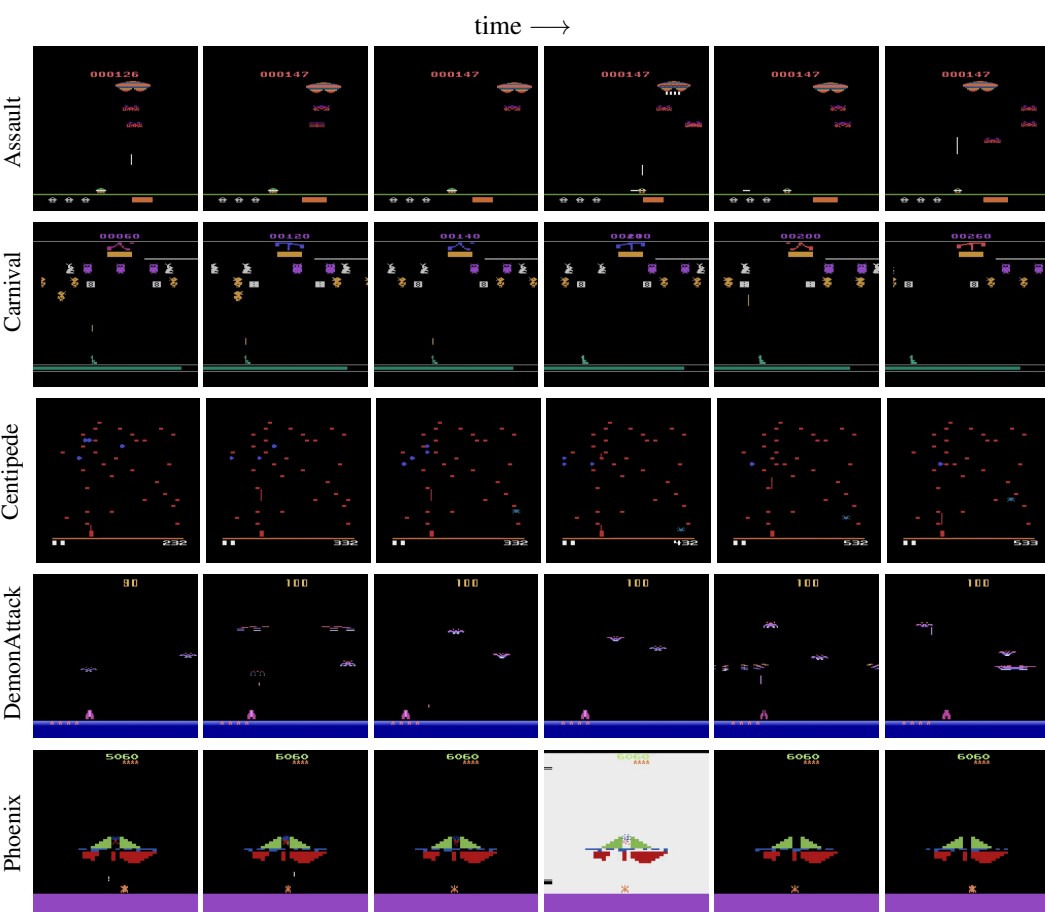

*Figure 16.* **Visualization of trajectories from the *Shooter* game category.** We visualize key frames in sample trajectories for five tasks. Actual trajectory lengths vary greatly between games.

time ⟶

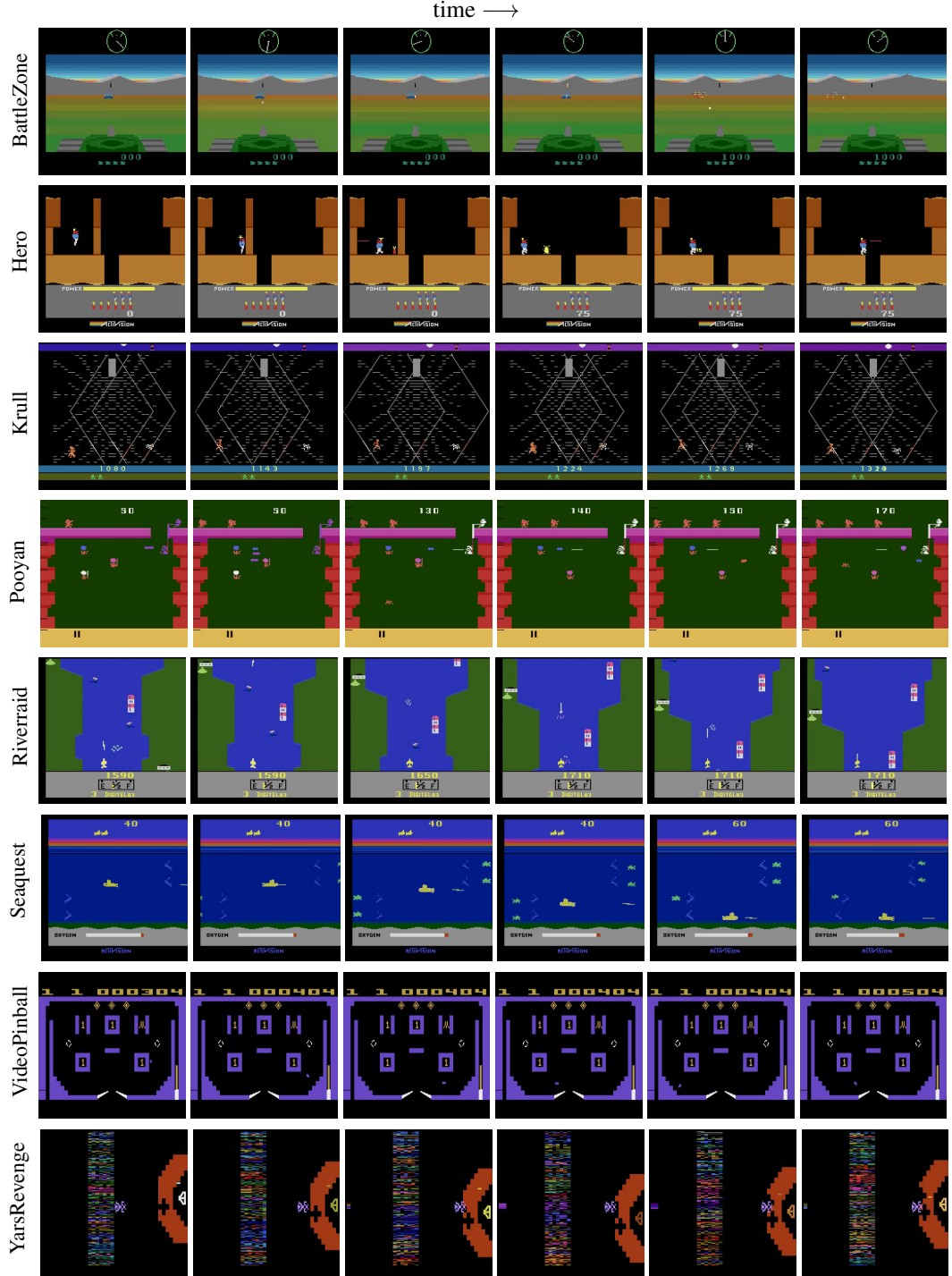

*Figure 17.* **Visualization of trajectories from the *diverse* game category.** We visualize key frames in sample trajectories for eight tasks. Actual trajectory lengths vary greatly between games.

