# OpenReview forum: "On the Feasibility of Cross-Task Transfer with Model-Based Reinforcement Learning"
_ICLR.cc/2023/Conference — ICLR 2023 poster_

### Official Review · Reviewer_k2wV · 2022-10-24

**Confidence:** 3
**Correctness:** 4
**Technical Novelty And Significance:** 3
**Empirical Novelty And Significance:** 3
**Recommendation:** 6

**Clarity, Quality, Novelty And Reproducibility:**

- Clarity and Quality: The paper is easy to read, clear in its statement and technically sound. The experimental results are clearly reported with good use of marking and coloring schemes. The experiments are repeated for multiple runs, and the aggregated metrics described in most of the captions.
- Novelty: the novelty of this work is difficult to evaluate because in practice, it combines existing techniques (finetuning on both online and offline datasets, gradient weighting based on gradient similarity). However, I acknowledge there is merit in studying the effectiveness of this approach on a diverse set of tasks.
- Reproducibility: the authors report the hyperparameters in the appendix but do not attach any code in the supplementary material, which makes its reproducibility only partially addressed.

**Strength And Weaknesses:**

The paper is well-written, and the problem under study is clearly defined. The proposed contribution in the loss formulation looks technically sound and the authors fully convey its rationale. Moreover, the experimental section shows an exemplary structure with several ablations that analyse different aspects of the proposed solution.

The main criticism is on the empirical evidence of the effect of gradient weighting. In the first experiment, the authors evaluate XTRA in tasks with similar game mechanics. While the performances are generally promising, one of the baselines uses constant weights and shows much higher performance in two games. While I acknowledge we cannot expect to have a solution for everything, I think the large performance gap disconfirms one of the main contributions of this work without properly commenting on it.

Moreover, even if focusing on model-based reinforcement learning, I would like to see the performance compared with model-free approaches. Even if the training budget is not comparable, reporting their asymptotic performance would give more context to the readers.

Minor errors:
- Missing reference to the original paper on World Models by Ha et al. (NeurIPS 2018).
- In Section 3.1, it is unclear what 'u' refers to when saying about "task specific quantities (pi, u, z)". From the appendix, 'u' is used for immediate reward, but I could not find its definition in the main text.
- Typo in 3.1: "Learning a single RL agent for a diverse set of tasks is however a difficult in practice" instead of "Learning a single RL agent for a diverse set of tasks is however difficult in practice".
- End of paragraph "Scaling model size": the sentence "XTRA is robust to model size and can be further benefit from scaling up the model size in the future" sounds a bit overselling and, in my opinion, it is not supported by sufficient experiments to claim something like that.
- Typo in Related work: "or example, He et al (2020)" instead of "For example, He et al (2020)"

**Summary Of The Paper:**

The paper studies the problem of cross-task transferability in model-based reinforcement learning by introducing a framework with offline pretraining of world models and online finetuning (XTRA).
XTRA builds on top of EfficientZero and introduces a novel weighted loss formulation to tackle two common problems: (1) catastrophic forgetting and (2) non-aligned downstream task with the training tasks. The experimental section compares the resulting approach with the original EfficientZero in several ablation studies, which highlight under which conditions the proposed solution shows positive results.
The authors conduct experiments on 14 visual tasks from the ALE benchmark and explicitly consider a reduced-budget setting.

**Summary Of The Review:**

Considering the lack of empirical support for the gradient-weighting mechanism, and the partial novelty in the overall framework, I vote for a weak rejection. By addressing these criticisms, the paper would be solid and mature for publication. I would recommend the authors to advance possible theses of the observed performance and validate them with additional experiments. For example, I would like to understand (1) to what extent the gradient weighting captures the task similarities and if the problem should be addressed by other mechanisms; and (2) if there is any model bias dependent on visual features that is playing a role in this performance gap.

---

> ### Author Response · Authors · 2022-11-17
> **Response to Reviewer k2wV**
>
> We thank the reviewer for their insightful comments. We address your comments in the following.
>
> ----
>
> **Q**: While the performances are generally promising, one of the baselines uses constant weights and shows much higher performance in two games. While I acknowledge we cannot expect to have a solution for everything, I think the large performance gap disconfirms one of the main contributions of this work without properly commenting on it. [...] The novelty of this work is difficult to evaluate because in practice, it combines existing techniques.
>
> **A**: Thank you for raising your concerns. We address your comment in two parts:
> 1) **Novelty**: We want to emphasize that our work is largely a **feasibility** study: we both contribute a new problem setting, an overall framework for tackling the problem, as well as solutions to specific challenges that arise in such a multi-task pretraining + online finetuning setting. This is acknowledged by reviewer MLsj: *“The paper tackles an important new problem that is of interest to the community.”*. We believe that our contributions are both substantial, timely, and pertinent to the community, due to its rapidly growing interest in pretraining and multi-task RL.
>
> 2) **Performance & ablation study**: We want to preface our answer by emphasizing that “*The authors tackle a notoriously hard problem: transfer across different tasks in RL, from images. They also chose one of the most competitive benchmarks to evaluate their method, namely Atari 100k, and include the most competitive baselines. In that sense, the results are impressive*” (reviewer bd3b). In this context, **we do believe that our results are both significant and somewhat surprising, and – given that our emphasis is on the feasibility of the problem setting itself – we do in fact consider the ablation w/ constant weights a contribution of our work as well**. To answer your question: we find that the two games for which constant weights outperform tasks weights are games that have a particularly great overlap in visuals, mechanics, and controls with the pretraining tasks, and retaining a task weight of 1 rather than downweighting can therefore be beneficial in this setting. For example, see Figure 5 (top row). In general, applying constant task weights is likely to be suboptimal if the relevance between pretraining games and target games is unknown. We are committed to including additional ablations on XTRA w/o gradient weighting in the diverse task setting for the camera-ready version.
>
> ----
>
> **Q**: Moreover, even if focusing on model-based reinforcement learning, I would like to see the performance compared with model-free approaches.
>
> **A**:Thanks for the suggestion. We have included **a comparison to CURL** (Srinivas et al., 2020), a strong model-free RL method – under the offline pretraining + online finetuning setting – in Table 2 of the revised paper.  We choose to experiment with CURL as it achieves better results on Atari100k than other model-free methods such as DrQ (the authors did not benchmark DrQ-v2 on Atari). Please refer to our general comment for experiment details. **From these preliminary results, we find that pretraining does not improve performance of the model-free method CURL as consistently as for our model-based framework, XTRA.** This result is supported by our finding that pretraining the latent dynamics model has the biggest influence on sample-efficiency in cross-task transfer (Figure 8). We are committed to including a larger set of experiments with CURL in the camera-ready version.
>
> ----
>
> We would also like to thank you for your remaining (minor) comments – we have addressed these in our revised manuscript. Please do not hesitate to let us know if you have any additional comments.

---

> > ### Comment · Reviewer_k2wV · 2022-11-21
> > **Response to the authors**
> >
> > W thank the authors for the helpful reply. It addresses most of the reviewers' comments and helps us to better assess the overall contribution. We also appreciate the commitment to including more ablations on gradient weighting, look forward to reading it. We will update our scores accordingly.

---

### Official Review · Reviewer_bd3b · 2022-10-24

**Confidence:** 4
**Correctness:** 3
**Technical Novelty And Significance:** 3
**Empirical Novelty And Significance:** 4
**Recommendation:** 6

**Clarity, Quality, Novelty And Reproducibility:**

- Clarity: mostly clear; my main qualm is about some details postponed to the Appendix (full explanation for the task reweighting scheme), whereas they could have been included with more careful writing.
- Quality: good; I’m not sure what Fig. 4 brings to the exposition but it doesn’t hurt.
- Novelty: good; the method is not so novel (tweaks on top of EfficientZero) but the transfer learning setting in RL is.
- Reproducibility: ok?; the authors promise to open-source their code but I don’t see a supplementary material.

**Strength And Weaknesses:**

- Strengths:
    - The authors tackle a notoriously hard problem: transfer across different tasks in RL, from images. They also chose one of the most competitive benchmarks to evaluate their method, namely Atari 100k, and include the most competitive baselines. In that sense, the results are impressive and XTRA is indeed successful at leveraging pretraining tasks to more efficiently solve the downstream one (my concerns on this point below).
    - To their credit, the authors also include some ablations studies and analyses (see 2nd part of 4.1) where they discuss task relevance, model size, and, more interestingly, which component of the model-based architecture is more transferrable and when. (Transferring the dynamics yields in the largest boost in the early finetuning phases.) I only wished those questions were explored in more depth, to help guide the community understand the mechanics of transfer in model-based RL on Atari.
- Weaknesses:
    - Since most of the experiments focus on benchmarking, the merits of the paper rest on XTRA’s performance. While it shows promising results, I also wonder if the results are as positive as described in the paper. First, how much more data than EfficientZero does XTRA require when taking pretraining into account (I couldn’t find those numbers in the main text)? Second, the confidence intervals are quite large when transferring to similar tasks (see Fig. 6: Carnival, DemonAttack, Alien, Amidar, Bankheist, MsPacman, Wizard of Wor) — are the reported results significant? It’s also surprising that XTRA does better relative to EfficientZero when downstream tasks are dissimilar from pretraining (see Table 2, confidence intervals) — could the authors comment on this point, and describe how the confidence intervals are computed in this setting?

        Depending on the answer to those questions, I wonder if this paper isn’t reporting negative results for cross-task RL transfer.

    - A more minor point: I wish the paper included better baselines — not just ablations of the proposed method. For example, how does XTRA perform against on-policy alternatives such as Actor-Mimic or Policy Distillation (e.g., on top of DrQ-v2)? In a similar vein, the authors claim their task-weighting scheme yields the same benefits as gradient surgery (see 3.2) but when is that empirically demonstrated?
    - One question re “Most improvements happen within the first  stages of training.”: Do the authors have any insight on why that’s the case? From Fig. 8 it seems that the dynamics are the reason for this boost (but we’re missing “h+f” on this figure). Is it because the dynamics already have a good initialization (i.e., don’t move too much during finetuning) or is it because the initialization lets them quickly move through parameter space (i.e., large movement at first, but not much after)?
    - Details:
        - What is the definition (formula) of $s^i$? I’m assuming it’s Eq. 2 but please clarify.
        - 3.1, p. 3: “tasks is however a difficult” → remove a.
        - 3.1, p. 4: you could cite Parisi et al.’s “Actor-Mimic”, and Rusu et al.’s “Policy Distillation”.
        - 3.1, p. 4: $(\hat{\pi}, \hat{u}, \hat{z})$ should be $(\hat{\pi}, \hat{v}, \hat{z})$?
        - 3.2, eq. 1: The sum should be $\sum_i^{m}$, right?
        - 4.1, p. 7: “our proposed frames” → “our proposed framework”?
        - Table 2, p. 7: \cite{} → \citep{} for Ye et al., 2021.
        - 4.1, p. 8: “This results indicates that (i) selecting […] and (2) in the […]” → enumeration consistency.

**Summary Of The Paper:**

This paper introduces XTRA, a model-based algorithm for reinforcement learning transfer. XTRA extends EfficientZero (SOTA model-based RL on Atari games) to the RL transfer setting in 2 steps:

1. Offline multi-task pretraining: distilling N EfficientZero teachers (pretrained on N tasks via offline RL) into a single EfficientZero student.
2. Online finetuning: where the student keeps optimizing for the mixture of pretraining tasks while also learning the downstream task via online RL.

Besides this pretraining-finetuning setup for model-based RL, the authors propose a scheme to reweight pretraining tasks according to their affinity with the downstream task — this aims to mimic the effects of gradient-surgery (Yu et al., 2020).

They evaluate XTRA on the Atari 100k benchmark (learning with only 100k frames), both when transferring between tasks that are similar (e.g., pretraining and downstream are instances of shooter tasks) and when they are dissimilar (i.e., not from any particular game family).

**Summary Of The Review:**

- Strengths:
    - The authors tackle a hard RL transfer problem, and show promising results.
    - Some ablation studies and analyses — I wish there were more.
- Weaknesses:
    - The reported results are not 100% compelling. I’ve asked the authors to clarify.
    - The paper could include better baselines, some of which need to be discussed as prior work (Actor-Mimic / Policy Distillation).

---

> ### Author Response · Authors · 2022-11-17
> **Response to Reviewer bd3b (1/2)**
>
> We thank the reviewer for their insightful comments. We address your comments in the following.
>
> ----
>
> **Q**: [...] how much more data than EfficientZero does XTRA require when taking pretraining into account (I couldn’t find those numbers in the main text)?
>
> **A**: First, we want to preface our answer by emphasizing that in practice, it is often costly to obtain data for a specific target task, but relatively easy to source diverse interaction data from elsewhere (e.g., public datasets, other tasks). In this paper, we investigate whether it is even **feasible** for model-based RL algorithms to benefit from offline multi-task data that are markedly different from the target task. Most of our experiments therefore focus on the relationship between finetuning performance, design choices, and the choice of pretraining tasks.
>
> To address your question, we include details on the offline dataset in Appendix F of our new revision. In summary, we collect 768 trajectories per game using EfficientZero (Ye et al., 2021) checkpoints of varying quality (from 0 to 120k training steps on Atari100k). This translates to an average of 1M transitions per game, but varies depending on episode length – for example, this only results in 636k transitions for the game of Assault. Thus, we show that XTRA is effective even when pretraining data is suboptimal, ultimately allowing us to learn from very diverse data sources.
>
> ----
>
> **Q**: It’s also surprising that XTRA does better relative to EfficientZero when downstream tasks are dissimilar from pretraining (see Table 2, confidence intervals) — could the authors comment on this point, and describe how the confidence intervals are computed in this setting?
>
> **A**: When pretraining tasks are sufficiently diverse, it is likely that some of the tasks will be useful for learning the target task. Further, we increase the number of pretrained tasks from 4 to 8 in the diverse setting (Table 2) compared to the Table 1 results. We include task weights’ visualizations for the diverse task setting in Figure 9 (Appendix) of our revised manuscript. We observe that most (but not all) pretrained tasks contribute positively (i.e., have a non-zero task weight) during finetuning, which may help explain this phenomenon. Lastly, we want to emphasize that maze-to-shooter experiments can be viewed as controlled experiments with negative results, where we remove all (explicit) relationships between pretraining and target tasks – and indeed pretraining in this case is not very helpful to improve target task performance. We believe that these insights will be valuable for the community.
>
> Regarding confidence intervals: the confidence intervals are 95% bootstrap confidence intervals (1,000 samples) across 5 model seeds and 32 evaluation episodes.
>
> ----
>
> **Q**: A more minor point: I wish the paper included better baselines — not just ablations of the proposed method. For example, how does XTRA perform against on-policy alternatives such as Actor-Mimic or Policy Distillation (e.g., on top of DrQ-v2)?
>
> **A**: We have included a **comparison to CURL** (Srinivas et al., 2020), a strong model-free RL method – under the offline pretraining + online finetuning setting – in Table 2 of the revised paper.  We choose to experiment with CURL as it achieves better results on Atari100k than other model-free methods such as DrQ (the authors did not benchmark DrQ-v2 on Atari). Please refer to our general comment for experiment details. **From these preliminary results, we find that pretraining does not improve performance of the model-free method CURL as consistently as for our model-based framework, XTRA.** This result is supported by our finding that pretraining the latent dynamics model has the biggest influence on sample-efficiency in cross-task transfer (Figure 8). We are committed to including a larger set of experiments with CURL in the camera-ready version.

---

> > ### Author Response · Authors · 2022-11-17
> > **Response to Reviewer bd3b (2/2)**
> >
> > **Q**: In a similar vein, the authors claim their task-weighting scheme yields the same benefits as gradient surgery (see 3.2) but when is that empirically demonstrated?
> >
> > **A**: Our work XTRA studies how dynamically re-weighting pretraining tasks can benefit a single target task. This separates our work from gradient surgery designed for multi-task supervised/RL problems where objectives from each task are treated as equally important. Concretely, both works measure task gradient cosine similarity, but have fundamentally different motivations and goals. In gradient surgery, gradient cosine similarity is a threshold to determine when to perform gradient projection, and gradient projection is a solution (surgery) to compute non-conflicting gradients across all tasks. In XTRA, gradient cosine similarity is used as task weights to prevent gradient contributions from multiple offline tasks from conflicting with a single target task. In Table 1, we show that XTRA outperforms XTRA w/o task weights across all tasks except Demon Attack and Phoenix, validating its significance.
> >
> > ----
> >
> > **Q**: One question re “Most improvements happen within the first stages of training.”: Do the authors have any insight on why that’s the case?
> >
> > **A**: Yes! From Table 1, we observe that pretraining with XTRA enables reasonable zero-shot performance on certain games (e.g. Phoenix). This indicates that when a target game is sufficiently similar to a pretraining game, our model tends to transfer quite well. Based on this observation, we conjecture that – even when the games are less similar – pretraining provides a good network initialization for quickly finetuning to a target task.
> >
> > ----
> >
> > We would also like to thank you for your remaining (minor) comments – we have addressed these in our revised manuscript. Please do not hesitate to let us know if you have any additional comments.

---

### Official Review · Reviewer_MLsj · 2022-10-25

**Confidence:** 4
**Correctness:** 4
**Technical Novelty And Significance:** 3
**Empirical Novelty And Significance:** 4
**Recommendation:** 6

**Clarity, Quality, Novelty And Reproducibility:**

- Clarity
    - The paper is well written and organized. It clearly explains the motivation behind its design choices. The illustrations are also of high quality.
- Quality
    - The method makes sense. The experiments convincingly show that the proposed XTRA outperforms online RL that is trained from scratch without access to offline data. It would be nice to have additional comparisons as mentioned in Main Weaknesses.
- Novelty
    - This appears to be the first paper tackling the problem of offline pretraining a world model on multiple tasks and online finetuning on a different task.
- Reproducibility
    - Experiments are run with 5 seeds. Architecture details and hyperparameters are provided in Appendix. The authors "are committed to" releasing the code. So it seems likely to reproduce the results.

**Strength And Weaknesses:**

- Strengths
    - The paper tackles an important new problem that is of interest to the community. Pretraining on multiple tasks and then finetuning on a different target task is more realistic than pretraining and finetuning on the same task as done in previous work, and has the potential for better data efficiency.
    - The paper is well written and easy to follow. The method is well motivated.
    - The ablation and analysis are quite informative, answering interesting questions such as which components are beneficial to finetune and which stage of training does the proposed model bring the most improvement.
- Main Weaknesses
    - The baselines are mainly online RL methods which do not have access to offline data. The paper can be strengthened by comparing to methods that also do offline pretraining and online finetuning. For example, it is mentioned in related work that Online Decision Transformer (ODT) pretrains and finetunes on the same task. The paper can be much stronger if it shows that when pretrained on the target task, XTRA acheives similar performance as ODT, and when including other pretraining tasks, it outperforms ODT.
    - Although model-based RL enjoys high sample efficiency in online single-task setting, it seems unclear whether it makes sense or is beneficial to use model-based RL in this paper's setting (as opposed to model-free RL). I would expect different tasks to have very different dynamics, so that it is not only difficult to learn a single model for all pretraining tasks, but it is also difficult to transfer the learned model to the target task. A comparison to model-free online finetuning would help clarify.
- Questions and Minor Issues
    - Can you elaborate on what is catastrophic forgetting in online finetuning? This can make the paper more self-contained and help the reader better understand why you want to retain the offline data.
    - There is not much ablation in the diverse task setting. It would be interesting to see if pretraining and/or retaining offline data helps, and if the task weights are close to zero.
    - I am curious why you use distillation during pretraining but switch back to the EfficientZero loss during finetuning (Eq 1). Is it possible to still use the distillation loss in finetuing?
    - It seems the improvement over EfficientZero is even larger in the diverse task setting than in the similar task setting. Can you comment on that? Also, why is the maze-to-shooter result much worse than the diverse task setting?
    - Figure 5 seems not mentioned in the main text.

**Summary Of The Paper:**

The paper proposes XTRA, a framework for finetuning a model-based RL agent that is pretrained on offline datasets to a target task through online interactions. Unlike previous work that uses the same task for offline pretraining and online finetuning, this paper tackles the setting where there are multiple pretraining tasks that are different from the target task. For multi-task offline pretraining, the paper shows that simply learning a single model for all tasks using RL cannot work. Hence, it proposes to first learn a teacher model for each task, and then distill all teacher models into a single student model. For online finetuning, to overcome catastrophic forgetting, the paper proposes to use offline data together with online data. To minimize interference from irrelevant tasks in the offline data, the paper further proposes a re-weighting technique based on task similarity. The experiments focus on the Atari 100k benchmark, with EfficientZero being the backbone and main baseline. There are two settings: (1) pretraining on games that are similar to the target game, and (2) pretraining on diverse games. XTRA outperforms EfficientZero in both settings. Further analysis shows that it is crucial to finetune both the pretrained representation and the dynamics function, and that XTRA has a large advantage over EfficientZero in the early stage of training.

**Summary Of The Review:**

I am leaning toward acceptance. The paper makes a first attempt at multi-task offline pretraining and online finetuning using a world model. While additional comparisons to single-task / model-free settings are nice to have, the paper already provides sufficient new knowledge to the community.

---

> ### Author Response · Authors · 2022-11-17
> **Response to Reviewer MLsj (1/2)**
>
> We thank the reviewer for their insightful comments. We address your comments in the following.
>
> ----
>
> **Q**: The paper can be strengthened by comparing to methods that also do offline pretraining and online finetuning. For example, it is mentioned in related work that Online Decision Transformer (ODT) pretrains and finetunes on the same task. The paper can be much stronger if it shows that when pretrained on the target task, XTRA acheives similar performance as ODT, and when including other pretraining tasks, it outperforms ODT.
>
> **A**:
> Thanks for the great suggestion. We agree that it is informative to compare to other methods that do offline pretraining + online finetuning. Given that the ODT authors do not perform experiments on Atari games, we have opted to instead compare to a strong model-free RL method for which we have official numbers and a public implementation. We believe that this experiment is in a similar spirit to what the reviewer suggests. See next comment for a more detailed response on this.
>
> ----
>
> **Q**: Although model-based RL enjoys high sample efficiency in online single-task setting, it seems unclear whether it makes sense or is beneficial to use model-based RL in this paper's setting (as opposed to model-free RL). I would expect different tasks to have very different dynamics, so that it is not only difficult to learn a single model for all pretraining tasks, but it is also difficult to transfer the learned model to the target task. A comparison to model-free online finetuning would help clarify.
>
> **A**: We address your comment in two parts:
> 1) **Model-free RL baseline**. We have included **a comparison to CURL** (Srinivas et al., 2020), a strong model-free RL method – under the offline pretraining + online finetuning setting – in Table 2. Please refer to our general comment for experimental details. **From these preliminary results, we find that improvements from model-free RL (CURL with offline pretraining + online finetuning) are not as big as for our proposed model-based RL (XTRA - ours).** We are committed to including a larger set of experiments with CURL in the camera-ready version.
>
> 2) **Transferring the learned model**. From Figure 8 (transferring model components) of our manuscript, we observe that transferring the learned latent dynamics actually accounts for the majority of the improvement in sample-efficiency on the target task. Transferring representation network alone (without learned world model) leads to sub-optimal performance. This is an important finding in our paper that different tasks with different dynamics can be useful in cross-task transfer to target task. We appreciate reviewer’s recognition on the difficulty of learning and transferring learned model in our XTRA setting.
>
> ----
>
> **Q**: Can you elaborate on what is catastrophic forgetting in online finetuning?
>
> **A**: We have revised our manuscript to provide more context on this in page 4, but also want to elaborate a bit here in our response. We observe that directly finetuning the pretrained model often erases useful inductive priors learned during pretraining, consequently leading to degraded performance on the target task; this phenomenon is known as catastrophic forgetting and is well documented in the literature. For example, we observe catastrophic forgetting in Phoenix and DemonAttack. From Table 1, we observe that XTRA w/ pretraining only (i.e., XTRA w/o cross-task finetuning) underperforms the EfficientZero baseline. Our XTRA results (w/ pretraining + cross-task finetuning) demonstrate that the knowledge from pretrained tasks continue to benefit the target task significantly even during finetuning.

---

> > ### Author Response · Authors · 2022-11-17
> > **Response to Reviewer MLsj (2/2)**
> >
> >
> > **Q**: There is not much ablation in the diverse task setting. It would be interesting to see if pretraining and/or retaining offline data helps, and if the task weights are close to zero.
> >
> > **A**: Thanks for the suggestion! We have included task weights’ visualizations in the diverse task setting in the revised manuscript. Please refer to Figure 9 (Appendix) for more details. We are committed to including more ablations in the diverse task setting for the camera-ready manuscript.
> >
> > ----
> >
> >
> > **Q**: I am curious why you use distillation during pretraining but switch back to the EfficientZero loss during finetuning (Eq 1). Is it possible to still use the distillation loss in finetuing?
> >
> > **A**: We have revised the manuscript (page 5) to clarify this. To answer your question: in terms of loss function, the distillation loss is the same as the EfficientZero loss, except that policy/value targets are predicted by the teacher model during distillation (supervised learning) rather than using the MuZero Reanalysis algorithm of EfficientZero (online RL). During online finetuning, we use distillation targets for offline pretrained games and MuZero Reanalysis to compute targets for the online target task for which we do not have a teacher.
> >
> > ----
> >
> >
> > **Q**: It seems the improvement over EfficientZero is even larger in the diverse task setting than in the similar task setting. Can you comment on that? Also, why is the maze-to-shooter result much worse than the diverse task setting?
> >
> > **A**: When pretraining tasks are sufficiently diverse, it is likely that some of the tasks will be useful for learning the target task. Further, we increase the number of pretrained tasks from 4 to 8 in the diverse setting (Table 2) compared to the Table 1 results. We include task weights’ visualizations for the diverse task setting in Figure 9 (Appendix) of our revised manuscript. We observe that most (but not all) pretrained tasks contribute positively (i.e., have a non-zero task weight) during finetuning, which may help explain this phenomenon. Lastly, we want to emphasize that maze-to-shooter experiments can be viewed as controlled experiments, where we remove all (explicit) relationships between pretraining and target tasks – and indeed pretraining in this case is not very helpful to improve target task performance. We believe that these insights will be valuable for the community.
> >
> > ----
> >
> > We would also like to thank you for your remaining (minor) comments – we have addressed these in our revised manuscript. Please do not hesitate to let us know if you have any additional comments.

---

### Official Review · Reviewer_8xCu · 2022-10-25

**Confidence:** 3
**Clarity, Quality, Novelty And Reproducibility:** The paper is well written and easy to…
**Correctness:** 3
**Technical Novelty And Significance:** 3
**Empirical Novelty And Significance:** 2
**Recommendation:** 6

**Strength And Weaknesses:**

### Strength
- The proposed question is quite interesting.
- The method is easy to follow.
- EfficientZero is a reasonable choice to study this question with offline pre-training.

### Weakness
- Major
    - Could the authors please provide more details about the offline data set used during the pre-training stage? How do you obtain the offline data set? What is the size of this offline data set? How are the quality and size affect the downstream performance?
    - Since the proposed model has an offline pre-training stage, the improvement in sample efficiency seems not significant compared with EfficientZero, as shown in Figure 6.
    - The BC baseline is too weak, there is no policy improvement stage such as the fine-tuning used for the proposed model.
- Minor
    - Figure 1 and figure 2 seemed not being referred anywhere in the paper.

**Summary Of The Paper:**

This paper studies the question of whether pre-trained model-based RL can be effectively fine-tuned to solve downstream tasks. To answer this question, during pre-training, this paper proposed to deploy a teacher-student framework: learn several sing-task teacher models via offline RL and distill these single-task policies into a multi-task student policy. Then, the multi-task student model is fine-tuned with the online target task. Offline data and gradient re-weighting are used to prevent catastrophic forgetting and performance degradation during fine-tuning.

**Summary Of The Review:**

This paper studies the question of whether pre-trained model-based RL can be effectively fine-tuned to solve downstream tasks.  However, the experiments lack important analysis of the effect of the offline data set.

---

> ### Author Response · Authors · 2022-11-17
> **Response to Reviewer 8xCu**
>
>
> We thank the reviewer for their insightful comments. We address your comments in the following.
>
> ----
>
> **Q**: Could the authors please provide more details about the offline data set used during the pre-training stage? How are the quality and size affect the downstream performance?
>
> **A**: Thank you – these are good questions. We want to preface our answer by reiterating that in practice, it is often costly to obtain data for a specific target task, but relatively easy to source diverse interaction data from elsewhere (e.g., public datasets, other tasks). In this paper, we investigate whether it is even possible for model-based RL algorithms to benefit from offline data collected by existing model checkpoints from tasks that are markedly different from the target task. Most of our experiments therefore focus on the relationship between finetuning performance and the choice of pretraining tasks. However, we agree that it would be informative to include experiments that vary the quality and size of the pretraining datasets. As a preliminary result, we note we can already observe (from Table 1 and Table 2) an improvement in Assault with more pretraining tasks; Assault improves from 1294.6 (with 4 pretraining tasks) to 1742.2 (with 8 pretraining tasks). We are committed to including such experiments in the camera-ready version.
>
> We also include information and statistics on the offline dataset in Appendix F of our new revision. We collect 768 trajectories per game using EfficientZero (Ye et al., 2021) checkpoints of varying quality (from 0 to 120k training steps on Atari100k). This translates to an average of 1M transitions per game, but varies depending on episode length – for example, this only results in 636k transitions for the game of Assault. Thus, we show that XTRA is effective even when pretraining data is suboptimal, allowing us to learn from very diverse data sources.
>
> ----
>
> **Q**: Since the proposed model has an offline pre-training stage, the improvement in sample efficiency seems not significant compared with EfficientZero, as shown in Figure 6.
>
> **A**: We want to emphasize that our study is a **feasibility** study. As reviewer (bd3b) mentions, “*The authors tackle a notoriously hard problem: transfer across different tasks in RL, from images. They also chose one of the most competitive benchmarks to evaluate their method, namely Atari 100k, and include the most competitive baselines. In that sense, the results are impressive*”. In this context, we do believe that our results are both significant and somewhat surprising. For example, we include results on two sets of experiments: shooter games and maze games (numbers are in Table 1). We find that the effectiveness of pretraining is task dependent, e.g., we observe a mean improvement in shooter games of +36%, and +11% in maze games. However, we do observe improvements in nearly all games.
>
> ----
>
> **Q**: The BC baseline is too weak, there is no policy improvement stage such as the fine-tuning used for the proposed model.
>
> **A**: Thanks for the suggestion. We have included **a comparison to CURL** (Srinivas et al., 2020), a strong model-free RL method (i.e., with policy improvement), under our offline pretraining + online finetuning setting in Table 2. Please also check the general comment for details. **From these preliminary results, we find that improvements from model-free RL (CURL with offline pretraining + online finetuning) are not as big as for our proposed model-based RL (XTRA - ours).** We are committed to including a larger set of experiments with CURL in the camera-ready version.
>
> Lastly, we also include an additional BC offline finetuning baseline in Table 1, where the pretrained BC method is finetuned on offline data for the target task (generated by the EfficientZero baseline). We find that finetuning BC with target-specific data underperforms both the EfficientZero baseline and XTRA.
>
> ### BC Results
>
> | Game          | EfficientZero | XTRA    | BC (pretrain+finetune) |
> |---------------|---------------|---------|------------------------|
> | Assault       | 1027.1        | 1294.6  | 838.4                  |
> | Carnival      | 3022.1        | 3860.9  | 1952.4                 |
> | Centipede     | 3322.7        | 5681.4  | 1814.1                 |
> | Demon Attack  | 11523.0       | 14140.9 | 825.5                  |
> | Phoenix       | 10954.9       | 14579.8 | 427.6                  |
> | Alien         | 695.0         | 954.8   | 152.9                  |
> | Amidar        | 109.7         | 90.2    | 25.5                   |
> | Bank Heist    | 246.1         | 304.9   | 178.8                  |
> | Ms Pacman     | 1281.4        | 1459.7  | 550.0                  |
> | Wizard Of Wor | 1033.1        | 985.0   | 163.8                  |
>
> ----
>
> We would also like to thank you for your remaining (minor) comments – we have addressed these in our revised manuscript. Please do not hesitate to let us know if you have any additional comments.

---

> > ### Comment · Reviewer_8xCu · 2022-11-18
> > **Response**
> >
> > Thank you for your additional experiment results. It addresses one of my major concerns. I would like to raise my score to 6.

---

### Author Response · Authors · 2022-11-17
**General Response to Reviewers and AC**

We thank all reviewers for their thoughtful comments and general appreciation. We have revised our manuscript based on your feedback, and responded to each of your individual comments. We list **major** changes to our manuscript below; these changes have also been highlighted (red text) in the new version.

First, however, we would like to address a recurring comment from reviewers: **the addition of a model-free baseline that performs pretraining + finetuning**. We agree that this would be a valuable addition to our paper. We now include a direct comparison to CURL, a strong model-free RL baseline, under the same offline pretraining + online finetuning setting as XTRA. We choose to experiment with CURL as it achieves better results on Atari100k than other model-free methods such as DrQ. We implement CURL w/ pretraining + finetuning by following the experimental setup as for our model-based method: **(1)** we first pretrain on the same 8 offline tasks as our method, and then **(2)** finetune to the target task with online RL for 100k environment steps. Results are shown below; all numbers are means of 5 seeds. Based on these experiments, **we find that pretraining does not improve performance of the model-free method CURL as consistently as for our model-based framework, XTRA**. This result is supported by our finding that pretraining the latent dynamics model has the biggest influence on sample-efficiency in cross-task transfer (Figure 8). We believe that the addition of a model-free RL finetuning experiment strengthens our paper, and thank the reviewers for the suggestion. We are committed to including additional model-free RL baseline numbers in the camera-ready version.



|                    | Model-Based RL | Model-Based RL | Model-Free RL | Model-Free RL              |
|--------------------|:--------------:|----------------|---------------|----------------------------|
| game               | EfficientZero  | XTRA(ours)     | CURL          | CURL-(pretrain+finetune)   |
| assault            | 1263.1         | 1742.2         | 600.6         | 588.6                      |
| battle_zone        | 13871.2        | 14631.25       | 14870.0       | 16450.0                    |
| hero               | 9315.9         | 10631.8        | 6279.3        | 6294.5                     |
| krull              | 5663.3         | 7735.8         | 4229.6        | 3472.8                     |
| seaquest           | 1100.2         | 749.5          | 384.5         | 385.5                      |
| Normed Mean        | 1.29           | 1.87           | 0.75          | 0.6                        |
| Normed Median      | 0.33           | 0.35           | 0.36          | 0.4                        |


### Major revision list
- Table 1: Added BC results under an offline pretraining + offline finetuning setting
- Table 2: Added CURL results under an offline pretraining + online finetuning setting
- Figure 9: Added XTRA task weights visualization under the diverse setting (Table 2)
- Appendix F: Added offline data preparation details
- Appendix J: Added behavior cloning implementation details

Again, we really thank the reviewers for their constructive feedback. We believe that your comments have been addressed in this revision, but will be happy to address any further comments from reviewers.

Best,

Authors

---

### Decision · Program_Chairs · 2023-01-20

**Decision:**

Accept: poster

**Justification For Why Not Higher Score:**

The paper is technically sound and the approach is quite interesting. I would have recommended it for spotlight or oral if they had studied more environments other than atari. Atari 100k may require a nuanced understanding and the broader ICLR audience might resonate more if there were multiple domains across different modalities or realism constraints.

**Justification For Why Not Lower Score:**

This is an important topic with an interesting solution. The use of learning a single world model and using it to transfer learn on a new target task is an open and timely problem to study. This paper clearly proposes a sound approach and should be accepted into ICLR. I believe several people will reference and build on top of these results for the Atari 100k suite and beyond.

**Metareview: Summary, Strengths And Weaknesses:**

This paper studies the long standing problem of data efficient RL. Pretraining is still not widely used in training RL agents and this paper takes small steps towards that direction. It proposes a framework for data efficient online RL by first learning a world model from N diverse tasks and then fine tuning this model on a target task with gradient re-weighting and N-task experience replays. To avoid catastrophic forgetting and task interference, two key mechanisms are proposed on top of the EfficientZero based architecture: 1) jointly train on all task data during target fine tuning and 2) adaptively re-weight gradient contributions in unsupervised manner based on similarity to target task.

I believe this paper tackles a critical problem in RL -- how to leverage and transfer a world model for online RL. More work needs to be done in this area and this paper proposes a sound and interesting take on this problem. The paper is well written and relatively easy to follow. The ablation and analysis are informative and makes it clear which components are beneficial to fine tune and what contributions bring performance improvement. They also provide a strong baseline and 25% improvement on a subset of the 100k Atari benchmark.

I also believe that this paper is good enough (assuming it incorporates all the suggestions made during the review process) for other authors to cite for comparisons (and potentially build upon). There were a lot of concerns that came up during the discussion period about the novelty of this approach but nothing major was articulated -- there are novel elements in this approach with sound experimental validation. One of the concerns was that the target task is not truly out of distribution and is still in the atari world -- but I think this is not a fair criticism. Studying generalization of games that are closely related but not fully OOD is an interesting problem in itself that deserves attention. Another concern was that the empirical improvements was 25% and not high enough -- I think the baselines are strong and the approach is interesting; so this type of an improvement along with the formulation makes this approach sound and interesting for other works to build upon.


**Note From Pc:**

if the above contains the word "oral" or "spotlight" please see: "oral" presentation means -> notable-top-5% and "spotlight" means -> notable-top-25%. As stated in our emails, we are disassociating presentation type from AC recommendations

**Summary Of Ac-Reviewer Meeting:**

One question that came up was if this was the first paper to study model-based multi-task transfer on the 100k atari domain. During the discussion the multi-game decision transformer was mentioned but this does not seem like a model-based method and does not predict future states.

All reviewers felt like this paper was novel and interesting but could have been executed better to make it a strong accept. There was consensus that it tackles a meaningful problem that hasn't been tackled before. Comments ranged from "result is quite surprising to me but not sure if there is a deep insight", "tasks are within atari and not totally different", "transferring plan / dynamics model than actual critic was important -- this is very interesting. i would have expected otherwise", "quite difficult to learn single world model for all Atari tasks. even if they are all games, their vision inputs are different. in dreamer they learnt a separate world model. but this paper distills a single world model -- this is also interesting and a contribution.", "very interesting and ultimately the most important problem. did reasonable approach. they have generalization that is interesting. not wow enough to fight.".

So my understanding is that all reviewers find this work interesting but they didn't fight hard enough because this was shown on Atari. I personally think Atari 100k is interesting and deserves further exploration. The problem of model based transfer on unbounded domains is a completely open ended question -- it is not fair to judge the contribution under this lens.